# Cold-induced hyperphagia requires AgRP neuron activation in mice

Jennifer D Deem[1], Chelsea L Faber[1], Christian Pedersen[2], Bao Anh Phan[1], Sarah A Larsen[1], Kayoko Ogimoto[1], Jarrell T Nelson[1], Vincent Damian[1], Megan A Tran[1], Richard D Palmiter[3], Karl J Kaiyala[4], Jarrad M Scarlett[1,5], Michael R Bruchas[6,7,8], Michael W Schwartz[1], Gregory J Morton[1]*

[1]UW Medicine Diabetes Institute, Department of Medicine, University of Washington, Seattle, United States; [2]Department of Bioengineering, University of Washington, Seattle, United States; [3]Department of Biochemistry, Howard Hughes Medical Institute, University of Washington, Seattle, United States; [4]Department of Oral Health Sciences, School of Dentistry, University of Washington, Seattle, United States; [5]Department of Pediatric Gastroenterology and Hepatology, Seattle Children's Hospital, Seattle, United States; [6]Department of Anesthesiology and Pain Medicine, University of Washington, Seattle, United States; [7]Department of Pharmacology, University of Washington, Seattle, United States; [8]Center for the Neurobiology of Addiction, Pain, and Emotion, University of Washington, Seattle, United States

**Abstract** To maintain energy homeostasis during cold exposure, the increased energy demands of thermogenesis must be counterbalanced by increased energy intake. To investigate the neurobiological mechanisms underlying this cold-induced hyperphagia, we asked whether agouti-related peptide (AgRP) neurons are activated when animals are placed in a cold environment and, if so, whether this response is required for the associated hyperphagia. We report that AgRP neuron activation occurs rapidly upon acute cold exposure, as do increases of both energy expenditure and energy intake, suggesting the mere perception of cold is sufficient to engage each of these responses. We further report that silencing of AgRP neurons selectively blocks the effect of cold exposure to increase food intake but has no effect on energy expenditure. Together, these findings establish a physiologically important role for AgRP neurons in the hyperphagic response to cold exposure.

*For correspondence:
gjmorton@uw.edu

Competing interests: The authors declare that no competing interests exist.

## Introduction

In homeothermic species, maintenance of euthermia in the face of a wide range of ambient temperatures is critical for survival. In small homeotherms, countering cold stress requires not only that adaptive adjustments of heat production occur rapidly and potently, but also that they are achieved without depleting body fuel stored in the form of fat (*Gordon, 1993*). Thus, when animals are housed in a cool environment, energy expenditure swiftly increases to generate the heat needed to maintain core body temperature (*Cannon and Nedergaard, 2011*) and, so long as a food is readily available, a compensatory hyperphagia prevents changes to body fat mass (*Kaiyala et al., 2012*). While much is known regarding the neurocircuitry underlying cold-induced thermogenesis (*Madden and Morrison, 2019*; *Nakamura and Morrison, 2011*; *Tan and Knight, 2018*), the origins of cold-induced hyperphagia remain poorly understood.

One potential explanation for cold-induced hyperphagia is that it is mounted as a secondary response to the negative energy balance state that results from cold-induced thermogenesis,

analogous to what occurs in other states of negative energy balance. During caloric restriction, for example, the imposed negative energy balance and the associated reduction of body fat stores drives adaptive homeostatic responses aimed at returning body fat stores to pre-intervention values. Reduced energy expenditure and increased hunger drive are major components of this adaptive response, and both are thought to be primarily driven by humoral signals indicative of energy deficiency (e.g., reductions of leptin and insulin) (*Schwartz et al., 2000*). It has, therefore, been reasonably assumed that the hyperphagic response to cold requires the same signals of energy deficiency, and this hypothesis is supported by studies that have been conducted over time periods sufficient to induce hormonal changes (*Bing et al., 1998*; *Hardie et al., 1996*; *Puerta et al., 2002*).

Among central targets of these humoral feedback signals are neurons located in the hypothalamic arcuate nucleus (ARC) that express agouti-related peptide (AgRP) (*Hahn et al., 1998*) which, when activated, potently stimulate feeding (*Aponte et al., 2011*; *Atasoy et al., 2012*; *Krashes et al., 2011*). However, recent evidence suggests that, although AgRP neurons are regulated by humoral signals (*Hahn et al., 1998*; *Schwartz et al., 2000*), they may also be under feed-forward control (*Chen and Knight, 2016*; *Lowell, 2019*) such that they can be activated by neurocircuits that integrate sensory cues from the environment (*Betley et al., 2015*; *Chen et al., 2015*; *Mandelblat-Cerf et al., 2015*; *Zimmer et al., 2019*) and thereby avert a negative energy state. Based on these observations, we hypothesized a role for AgRP neuron activation in the adaptive increase in food intake induced by cold exposure, and that this effect is not secondary to the associated increase in energy expenditure or a state of negative energy balance.

## Results

### Effect of chronic cold exposure on determinants of energy balance and hypothalamic neuropeptide expression

Our first goal was to confirm the previous findings regarding the effect of chronic cold exposure on energy homeostasis in normal mice. Using a mild, chronic cold-exposure paradigm (14°C for 5 days) and comparing outcomes to mice housed at room temperature (22°C) (*Figure 1A*), we found that, as expected (*Cannon and Nedergaard, 2004*; *Morrison et al., 2014*), core body temperature did not change significantly relative to controls housed at room temperature (*Figure 1B*), presumably owing to an associated increase in heat production (*Figure 1C*). This thermogenic response was accompanied by a proportionate increase in energy intake (*Kaiyala et al., 2015*; *Ravussin et al., 2014*; *Vallerand et al., 1986*; *Figure 1D*), such that neither body weight nor body fat mass changed significantly over the course of the study (*Figure 1E–G*). Moreover, neither respiratory quotient (RQ) nor ambulatory activity changed significantly (data not shown).

At the end of the 5-day study, animals were sacrificed and hypothalami were rapidly dissected for RNA isolation. To identify candidate hypothalamic mediators of this cold-induced hyperphagic response, we used quantitative real-time PCR (qRT-PCR) to measure select neuropeptide transcripts in isolated hypothalamic RNA. Despite our use of a mild cold-exposure paradigm, consistent with previous work using more severe cold-exposure paradigms (~4–5°C) (*McCarthy et al., 1993*; *Tang et al., 2009*), we found increased expression of orexigenic *Agrp* and *Npy* mRNA in animals housed at 14°C when compared to controls housed at room temperature, while expression of *Pomc* and *Pmch* remained unchanged (*Figure 1H*). Importantly, this evidence of enhanced *Agrp* mRNA expression cannot be attributed to reductions of either food intake or body fat stores, since the former was actually increased and the latter unchanged throughout the study period. The data are more consistent with a model in which AgRP neuron activation contributes to cold-induced hyperphagia.

### Energy expenditure and energy intake increase rapidly upon acute cold exposure

To further resolve the precise timing of the onset of feeding and metabolic responses to mild cold exposure, we commenced serial metabolic measurements starting when mice were placed into metabolic cages (to which they had been previously acclimated) that were housed inside environmental chambers that were either pre-cooled (14°C) or set at room temperature (22°C) (*Figure 2A*). Our findings show that, in response to the sudden change in ambient temperature, increases in heat

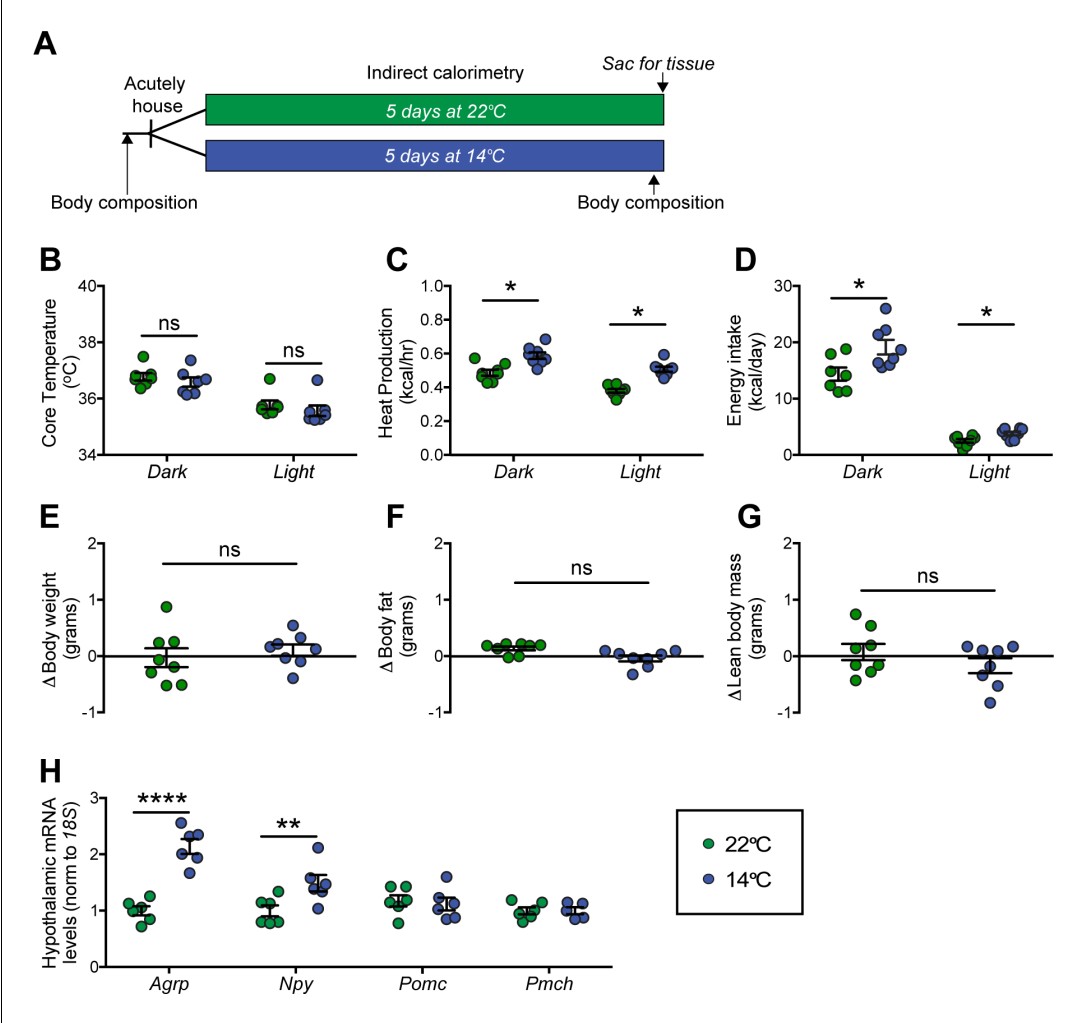

**Figure 1.** Effect of chronic cold exposure on determinants of energy balance and hypothalamic neuropeptide gene expression. (**A**) Adult male, wild-type mice were housed at the start of the dark cycle following body-composition measures and were maintained in temperature-controlled chambers set to either mild cold (14°C) or, as control, room temperature (22°C) for 5 days. Following a final body-composition analysis, animals were sacrificed and hypothalamic punches rapidly dissected. Mean dark and light cycle (**B**) core body temperature, (**C**) heat production, and (**D**) energy intake over 5 days while housed at either 22°C or 14°C. Change in (**E**) body weight, (**F**) body fat, and (**G**) lean body mass at study end in the same mice, n = 7–8 per group, mean ± SEM. Student's t-test, *p<0.05 vs. 22°C. (**H**) Hypothalamic mRNA levels of agouti-related peptide (*Agrp*), neuropeptide Y (*Npy*), pro-opiomelanocortin (*Pomc*), and pro-melanin concentrating hormone (*Pmch*) as determined by qRT-PCR, mean ± SEM. Two-way ANOVA with Holm-Sidak correction for multiple comparisons, *p<0.05, **p<0.01, ****p<0.0001 vs. 22°C.

The online version of this article includes the following source data for figure 1:

**Source data 1.** Wild-type chronic study.

production occurred rapidly, such that the thermogenic response needed to maintain core temperature was detected within 5 min (*Figure 2B*) and remained elevated for the duration of the 24 hr study (*Figure 2C*). Somewhat surprisingly, we also found that upon acute cold exposure, the increase in food intake required to offset heightened thermogenic demand was equally swift (*Figure 2D*). By comparison, neither RQ nor ambulatory activity was affected during cold exposure, suggesting that they are unlikely to play an important role in the adaptive response to this challenge (*Figure 2—figure supplement 1*). Like the increase in heat production, the initial increase in the rate of food consumption was maintained throughout both light and dark cycles (*Figure 2E*). Based on the rapidity of this feeding response, we infer that it was unlikely to have resulted from any detectable depletion of body fat or associated circulating signal (e.g., leptin). We interpret these findings as favoring a model in which the perception of cold is sufficient to increase both food intake and

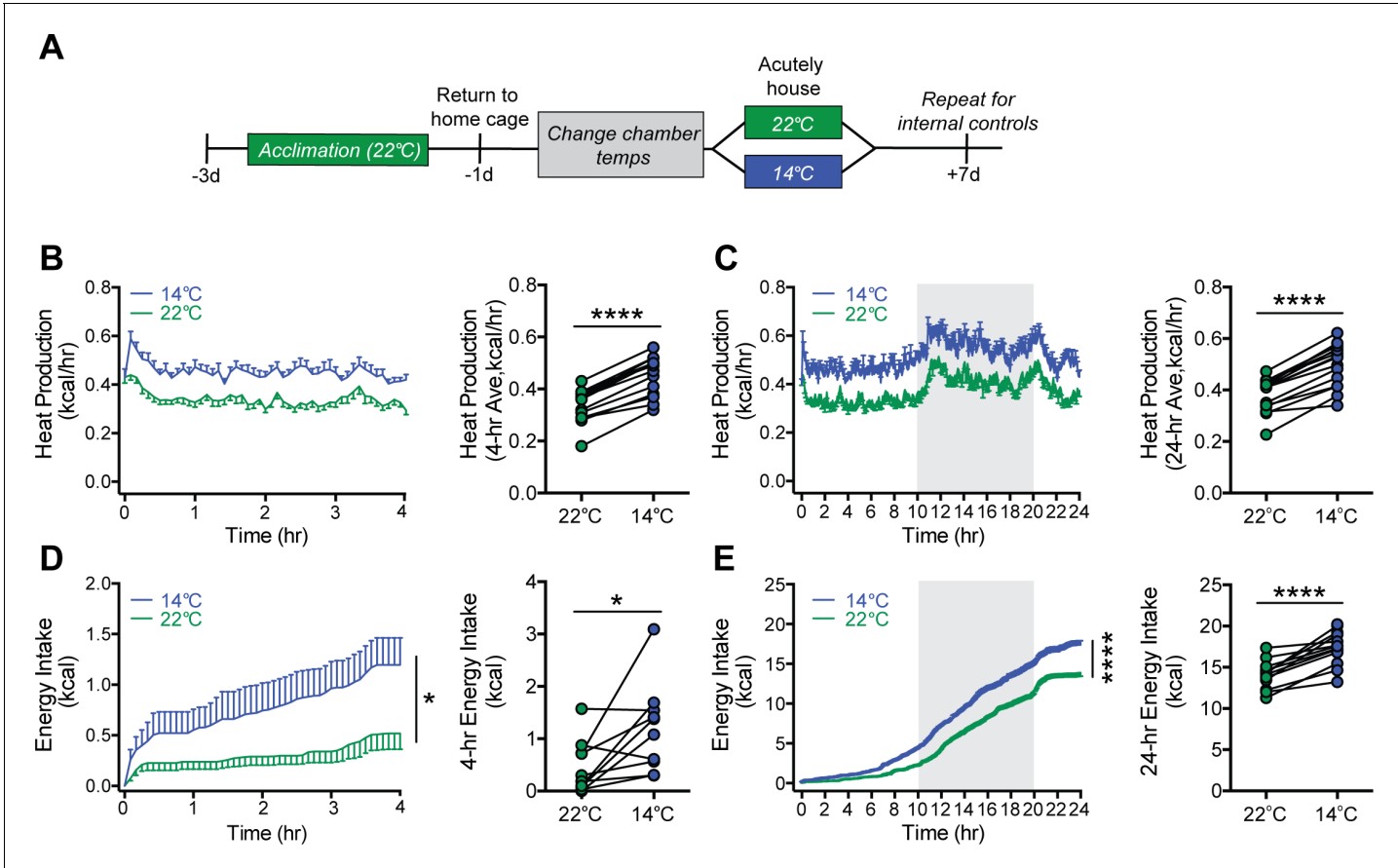

**Figure 2.** Acute mild cold exposure rapidly increases both energy expenditure and energy intake. (A) Adult male wild-type mice were acutely housed in temperature-controlled chambers set to either mild cold (14℃) or, as a control, room temperature (22℃). Mean heat production (B and C) and energy intake (D and E) were measured over either 4 hr (B and D) or 24 hr (C and E), beginning at 10 AM, t = 0. n = 10 per group, mean ± SEM. ****p<0.0001, *p<0.05, based on mixed factorial-ANOVA for changes of food intake over time, and paired Student's t-test to compare mean values of food intake and heat production.

The online version of this article includes the following source data and figure supplement(s) for figure 2:

**Source data 1.** Wild-type acute study.

**Figure supplement 1.** Effect of acute mild cold exposure on respiratory quotient (RQ) and ambulatory activity.

energy expenditure rapidly and in parallel, independently of feedback signals that might be subsequently recruited.

## Mild cold exposure induces Fos in AgRP neurons

We next investigated whether the rapid onset of cold-induced hyperphagia is associated with AgRP neuron activation. To this end, we utilized transgenic AgRP-Cre:GFP mice to enable visualization of Fos protein, a marker of neuronal activation (*Kovács, 1998*), in AgRP neurons after placement in cages previously set to mild cold (14℃), room temperature (22℃), or thermoneutrality (30℃) for 90 min with food present. We then quantified the total number of Agrp+ (GFP), Fos+, and Fos+/AgRP + across the entire rostral to caudal ARC (*Figure 3A*) with representative images from the center of this range presented in *Figure 3B*. Although the total number of AgRP+ neurons was comparable in all mice tested (*Figure 3C*), relative to mice maintained at 22℃, we found that both the total number of Fos+ cells in the ARC (*Figure 3D*) and, specifically, the number of Fos+ AgRP neurons (*Figure 3E and F*) was increased within 90 min of cold-exposure onset. However, while the percentage of AgRP neurons that expressed Fos in response to cold exposure (~25–30%) is similar to that previously reported in P10 mice separated from the warmth of the nest, dam, and littermates (*Zimmer et al., 2019*), it is less than that described following fasting (*Liu et al., 2012*). As expected,

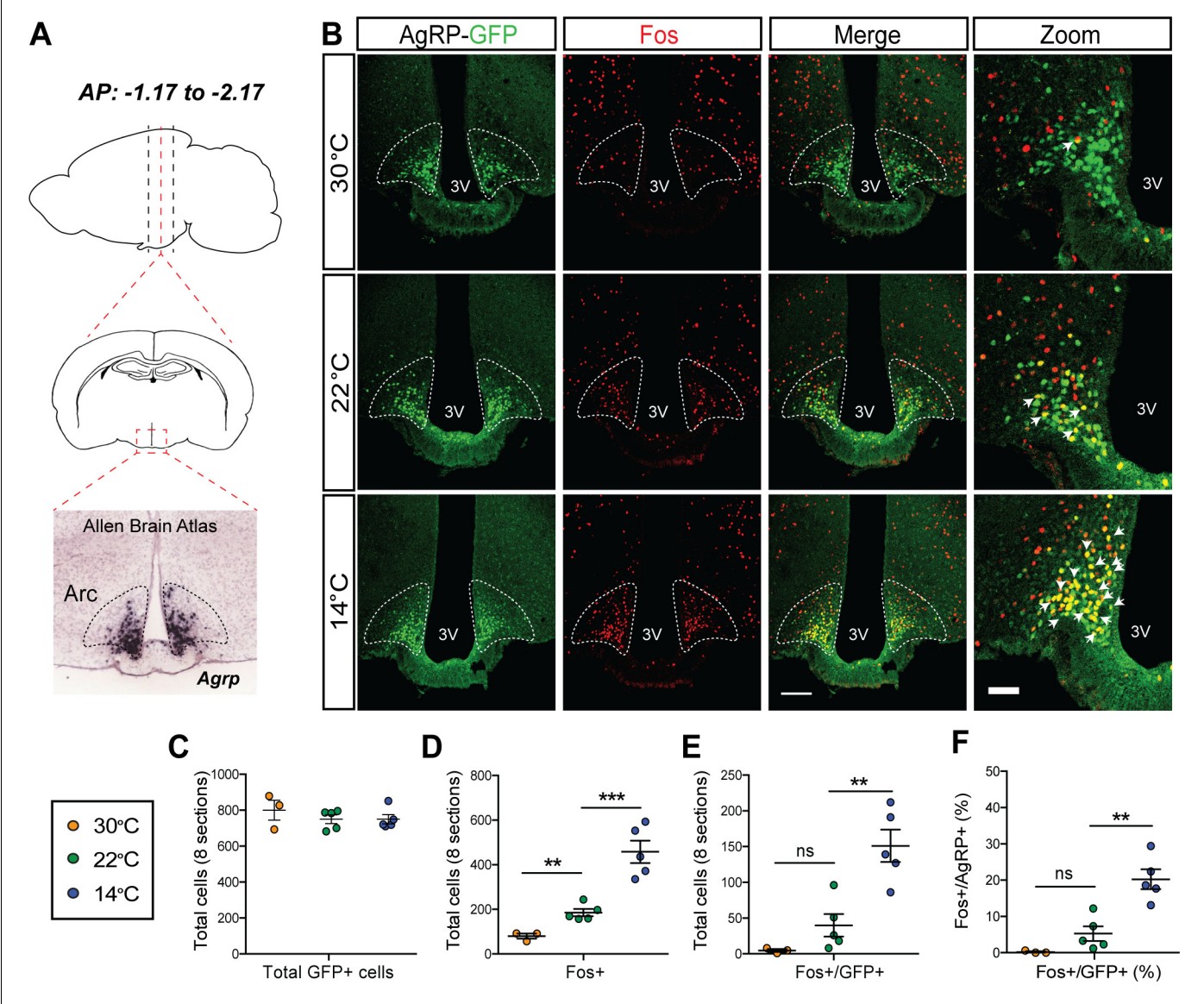

**Figure 3.** Acute mild cold exposure activates agouti-related peptide (AgRP) neurons. (A) Representative sagittal and coronal images and *Agrp* hybridization in situ from Allen Brain Institute of the arcuate nucleus (ARC). (B) Immunohistochemical detection of AgRP-locus driven green fluorescent protein (GFP) (green), Fos (red), and colocalization of GFP and Fos (two right panels) in the ARC of AgRP-Cre:GFP mice 90 min after housing at either 14°C, 22°C, or 30°C. Quantitation of (C) total AgRP+ cells, (D) total Fos+ cells, and the total number (E) and percent (F) of AgRP neurons that co-express Fos across eight rostral to caudal sections of the ARC (AP: −1.17 to −2.17). Thin bar = 100 µm, thick bar = 50 µm, n = 3–5group, mean ± SEM. One-way ANOVA with Sidak correction for multiple comparisons, ***p<0.001, **p<0.01, *p<0.05 vs. 22°C.

The online version of this article includes the following source data and figure supplement(s) for figure 3:

**Source data 1.** Arcuate nucleus Fos quantification.

**Figure supplement 1.** Fos induction in known thermoregulatory brain regions.

**Figure supplement 2.** Fos induction in agouti-related peptide (AgRP) neurons across entire arcuate nucleus (ARC).

we also found cold-induced increases in Fos in other thermoregulatory control centers including the parabrachial nucleus (PBN), preoptic area (POA) (*Bratincsák and Palkovits, 2004*; *Geerling et al., 2016*), dorsomedial hypothalamus (DMH) (*Hunt et al., 2010*), and rostral raphe pallidus (rRPa) (*Figure 3—figure supplement 1*). Moreover, in contrast to both 14°C and 22°C, both the total number of Fos+ cells in the ARC and the number of Fos+ AgRP neurons were minimal in mice housed at 30°C (*Figure 3D–F*). This Fos response was distributed equivalently across the entire rostral to caudal

AgRP neuron population (*Figure 3—figure supplement 2*). Our finding that Fos+ AgRP neuron number increases at temperatures below thermoneutrality (*Figure 3F*) identifies ambient temperature as a potential physiological regulator of AgRP neuron activity.

## Mild cold exposure rapidly increases AgRP neuron activity

To definitively establish the timing with which AgRP neuron activity changes in relationship to cold exposure, we applied fiber photometry to record calcium dynamics of AgRP neurons in vivo during a thermal challenge. To this end, AgRP-IRES-Cre mice received a unilateral injection of an adeno-associated virus (AAV) containing a Cre-dependent, genetically encoded calcium sensor GCaMP6s directed to the ARC (*Chen et al., 2013*), followed by implantation of an optical fiber at the injection site (*Figure 4A and B*). After allowing 3 weeks for transgene expression and acclimation, validation of signal quality was performed by examining the AgRP neuron response to food presentation in the fasted state. Consistent with previous observations (*Betley et al., 2015*; *Chen et al., 2015*; *Mandelblat-Cerf et al., 2015*), we found that food presentation strongly and rapidly inhibited AgRP neuron activity, as indicated by a strong reduction in GCaMP signal (*Figure 4—figure supplement 1*). To determine whether exposure to cold affects AgRP neuron activity, mice were placed in a custom-built plexiglass cage modified to enable rapid control over the temperature sensed by the tethered animal (*Figure 4C*). We predicted that if AgRP neurons contribute to cold-induced hyperphagia, their activation must precede or coincide with hyperphagia onset (*Figure 2D*). Consistent with this prediction, we found that when the temperature of the platform was rapidly reduced from 30°C to 14°C (within 1 min) an increase in GCaMP signal was detected within seconds, consistent with an increase in AgRP neuron activity. Moreover, this effect was reversed by raising the temperature from 14°C back to 30°C (*Figure 4D–F*).

To further validate the photometry signal, at the end of each study, animals were presented with a food pellet to ascertain intact inhibition of AgRP neuron GCaMP activity. Although the animals were fed ad libitum prior to the study, a clear inhibition of AgRP neuron GCaMP activity was present comparable to the changes seen during temperature transitions, thus validating the photometry signal (*Figure 4—figure supplement 2*). Taken together, these findings indicate that AgRP neurons are rapidly activated in response to cold exposure in a reversible manner.

## AgRP neuron activity is necessary for cold-induced hyperphagia

Having established that AgRP neurons are rapidly activated in response to mild cold exposure, we next sought to determine whether this response is required for the associated hyperphagia. To test this hypothesis, we utilized a chemogenetic approach in which AgRP-IRES-Cre mice received bilateral microinjections into the ARC of an AAV construct containing a Cre-dependent cassette encoding the inhibitory designer receptor activated by a designer drug (DREADD), hM4Di:eYFP (*Sanford et al., 2017*) or a fluorescent reporter control alone (mCherry) (*Figure 5A*). The transduced neurons can be detected by their eYFP expression and as expected, were limited to the ARC (*Figure 5B*). To further validate the ability of the inhibitory DREADD to reduce AgRP neuron responsiveness, we found that pretreatment injection of CNO in mice expressing the inhibitory DREADD in AgRP neurons attenuated cold-induced induction of Fos in AgRP neurons relative to saline-treated controls (*Figure 5—figure supplement 1*).

To determine whether activation of AgRP neurons is required for intact hyperphagic responses to cold exposure, acclimated animals received an intraperitoneal injection of clozapine-N-oxide (CNO) or its vehicle (saline) in a randomized crossover manner, 1 hr prior to being placed into metabolic cages housed within temperature-controlled chambers previously set at either 22°C, 30°C, or 14°C for measures of food intake and energy expenditure by indirect calorimetry for 4 hr (*Figure 5C*). Based on our Fos and fiber photometry data (*Figures 3* and *4*), we predicted that the effect of chemogenetic AgRP neuron inhibition would be robust in mice housed at 14°C, but minimal at 30°C, because AgRP neurons exhibit low basal activity at thermoneutrality. Consistent with this prediction, we found that whereas inhibition of AgRP neurons had little effect on food intake in mice housed at 30°C (*Figure 5D*), intake was strongly reduced in animals housed at 14°C, while intake was only modestly inhibited by CNO in mice housed at 22°C (*Figure 5F and H*). The latter finding is consistent with room temperature posing a thermal stress to mice (*Abreu-Vieira et al., 2015*). Moreover, these responses cannot be explained by a nonspecific effect of CNO, as cold-induced increases of food

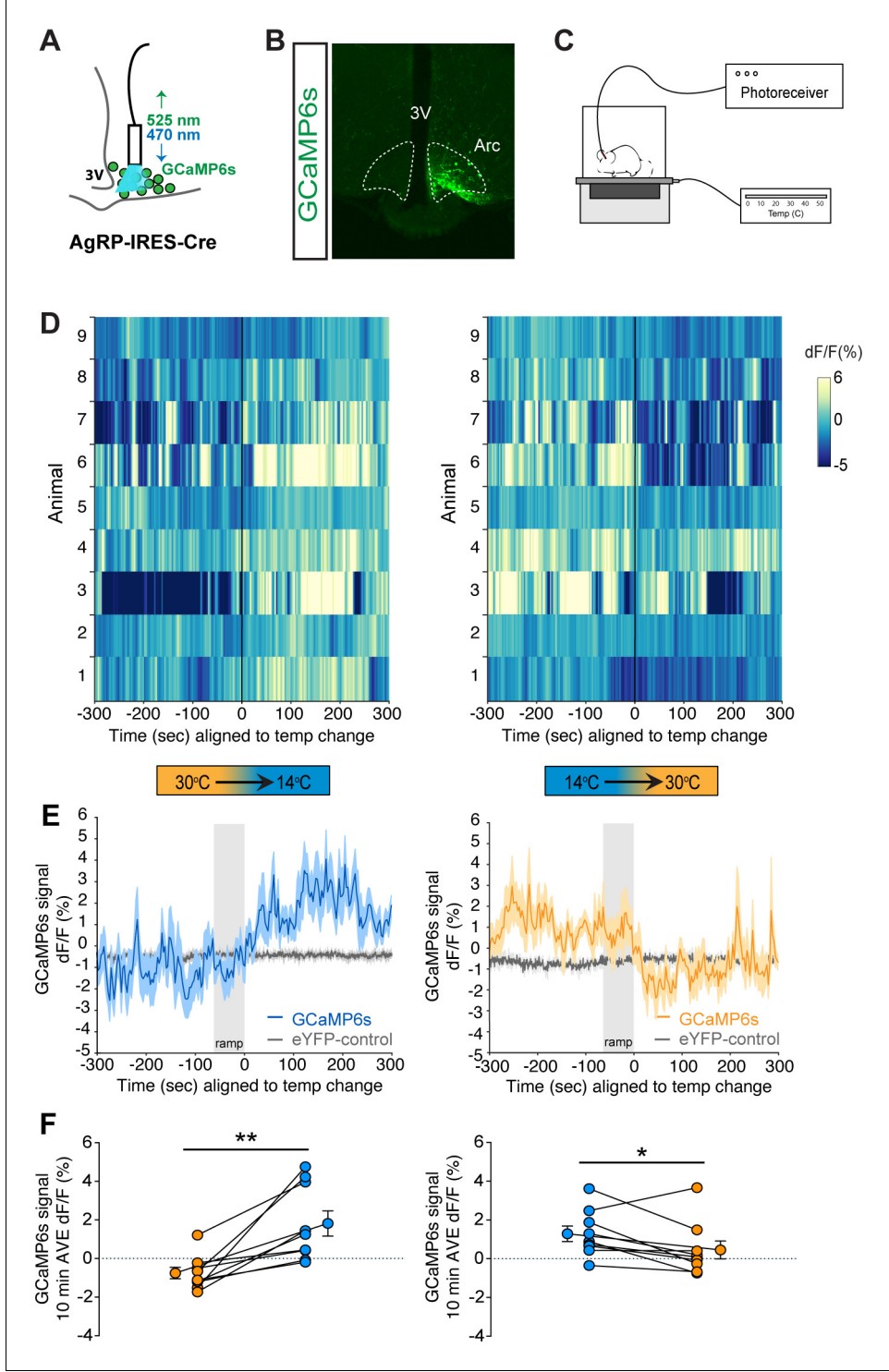

**Figure 4.** Cold sensation increases agouti-related peptide (AgRP)-neuron GCaMP activity in a rapidly reversible manner. (**A**) Representative diagram of fiber photometry with fiber placement at the arcuate nucleus. (**B**) Unilateral GCaMP6s expression in AgRP neurons. (**C**) Representative diagram of experimental set up. (**D**) Heat maps of dF/F (%) for individual animals, (**E**) trace of averaged dF/F (%) GCaMP6s signal, and (**F**) quantification of mean dF/F (%) differences between 30°C and 14°C GCaMP6s activity during 10 min exposure. Gray bar signifies 60 s temperature-ramp transition from either 14°C to 30°C or 30°C to 14°C, denoted by 'ramp', n = 9 per group, mean ± SEM. Student's paired t-test, **p<0.01, *p<0.05.

The online version of this article includes the following source data and figure supplement(s) for figure 4:

*Figure 4 continued on next page*

*Figure 4 continued*

**Source data 1.** Averaged photometry data.
**Figure supplement 1.** Post-fast refeeding inhibits agouti-related peptide (AgRP) neurons.
**Figure supplement 2.** Agouti-related peptide (AgRP) neuron GCaMP6s activity is reduced by food presentation in ad lib fed mice following thermal challenge.

intake and EE in AgRP-IRES-Cre mice expressing a control mCherry fluorophore did not differ following i.p. injection of CNO vs. saline (*Figure 5—figure supplement 2*).

The effect of AgRP neuron inhibition appears to be selective for energy intake, as the cold-induced increase in heat production was not affected (*Figure 5E,G, and I*). In addition, there was no significant effect of AgRP neuron inhibition on either RQ or ambulatory activity at any of the three ambient temperatures studied (*Figure 5—figure supplement 3*). Taken together, these findings suggest that AgRP neuron activation is required for cold-induced hyperphagia, and furthermore, distinct and separable neurocircuits underlie the feeding and thermogenic responses to cold. In addition, some degree of cold stress (including housing at room temperature) is required for food intake to be reduced by chemogenetic, AgRP neuron inhibition.

## Discussion

The tight coupling that exists between thermoregulation and energy homeostasis (*Gordon, 1993*; *Kaiyala et al., 2015*) enables animals to preserve both core body temperature and body fat mass across a wide range of ambient temperatures, so long as food is available (*Kaiyala et al., 2015*; *Smith and Romsos, 1984*; *Thurlby and Trayhurn, 1979*). Fundamental to this process is the precise coupling between changes of energy expenditure and energy intake in response to changing requirements for heat production (*Armitage et al., 1984*; *Kaiyala et al., 2015*; *Thurlby and Trayhurn, 1979*). While cold-induced hyperphagia has been documented across many species, including humans (*Johnson and Kark, 1947*), the mechanisms underlying this response have received little attention, especially when compared to the considerable literature describing how changes of thermogenesis are coupled with changes of ambient temperature (*Madden and Morrison, 2019*; *Rezai-Zadeh and Münzberg, 2013*). This relative lack of research interest may reflect the pervasive view that rather than being an integral component of the thermoregulatory control system, cold-induced hyperphagia is mounted as a compensatory response to the negative energy state induced by cold-induced thermogenesis. Our current data suggest that this perspective should be revised.

Given the well-documented role of AgRP neurons in physiological control of food intake and energy homeostasis (*Schwartz et al., 2000*; *Timper and Brüning, 2017*; *Varela and Horvath, 2012*), we hypothesized that cold-induced hyperphagia is driven at least in part by activation of these neurons. We report that in normal mice, AgRP neurons are rapidly activated by cold exposure and that intact cold-induced hyperphagia requires this activation, since increased food intake in this setting was blocked by acute chemogenetic AgRP neuron silencing. We also show that the magnitude of food intake reduction induced by chemogenetic silencing of these neurons varies inversely with increasing ambient temperature, such that the effect is more pronounced in colder environments relative to thermoneutral conditions in which AgRP neuron activity is lower. Taken together, we conclude that under our experimental conditions (e.g., in the absence of negative energy balance induced, for example, by caloric restriction), the adaptive increase in food intake during acute cold exposure depends upon increases in AgRP neuron activity.

AgRP neurons are among the most studied hypothalamic neuronal populations involved in energy homeostasis. These GABAergic neurons co-express the potent orexigen neuropeptide Y (NPY) and are found exclusively in the ARC (*Hahn et al., 1998*). An adjacent neuronal population expresses pro-opiomelanocortin (POMC), and whereas AgRP neuron activation stimulates feeding, POMC-neuron activation has the opposite effect (*Schwartz et al., 2000*; *Xu et al., 2011*). Furthermore, AgRP and POMC neurons are reciprocally regulated by leptin (*Schwartz et al., 2000*; *Sohn et al., 2013*) and other humoral signals, such that the effect of energy restriction to reduce body fat stores (and plasma leptin levels) causes both activation of AgRP neurons and inhibition of POMC neurons, a combined response that drives feeding until body fat stores and plasma leptin levels return to baseline values (*Schwartz et al., 2017*). These considerations raise the possibility that the feeding

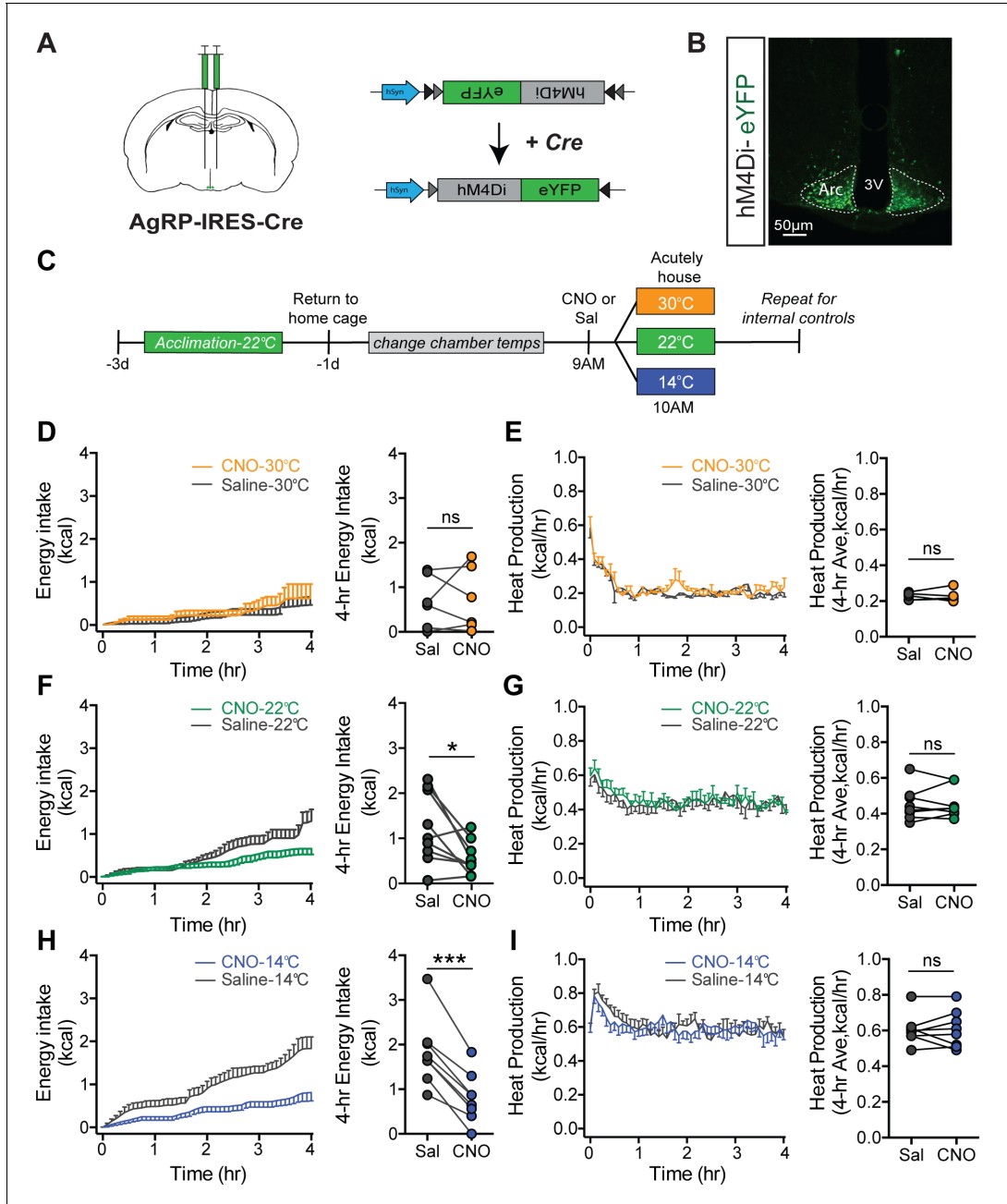

**Figure 5.** Cold-induced hyperphagia, but not thermogenesis, requires activation of agouti-related peptide (AgRP) neurons. (**A**) Schematic depicting strategy for bilateral microinjection of the Cre-dependent inhibitory DREADD (hM4Di) virus into the arcuate nucleus (ARC) of AgRP-IRES-Cre mice. (**B**) Detection of bilateral hM4Di-eYFP in transduced AgRP neurons in the ARC. (**C**) Adult male hM4Di-eYFP AgRP-IRES-Cre mice were acclimated to temperature-controlled chambers set to 22°C. Animals were returned to their home cages and chamber temperatures were adjusted overnight. In the morning, animals were dosed i.p. with either CNO or saline 1 hr prior to being acutely housed at either mild cold (14°C), room temperature (22°C), or thermoneutrality (30°C) in a randomized, crossover manner. (**D, F, and H**) 4 hr time-series and mean values for energy intake, and (**E, G, and I**) 4 hr time-series and mean values of heat production in hM4Di-eYFP AgRP-IRES-Cre mice that were housed at either 30°C, 22°C, or 14°C after receiving an i.p. injection of either saline or CNO n = 6–8 per group, mean ± SEM. RM-ANOVA and Student's t-test, ***p<0.001, *p<0.05 vs. saline.

The online version of this article includes the following source data and figure supplement(s) for figure 5:

**Source data 1.** Inhibition of agouti-related peptide neurons.
**Figure supplement 1.** Inhibition of cold-induced Fos in hM4Di-expressing agouti-related peptide (AgRP) neuron.
**Figure supplement 2.** CNO controls associated with inhibition of agouti-related peptide (AgRP) neurons during acute exposure to three ambient temperatures.
**Figure supplement 2—source data 1.** mCherry control food intake and heat production.
*Figure 5 continued on next page*

*Figure 5 continued*

**Figure supplement 3.** Respiratory quotient (RQ) and ambulatory activity after agouti-related peptide (AgRP) neuron inhibition at three ambient temperatures.

**Figure supplement 3—source data 1.** Respiratory quotient and activity.

response to cold is mounted as a compensatory response triggered by increased thermogenesis, which in turn causes depletion of body fuel stores and associated reduction of adiposity (e.g., falling plasma leptin levels) (*Bing et al., 1998*; *Puerta et al., 2002*; *Ricci et al., 2000*; *Tang et al., 2009*). However, our data suggests that the rapidity with which AgRP neurons are activated in response to cold exposure is unlikely to be explained by this type of negative feedback control. Nevertheless, humoral negative feedback signals could certainly be recruited over time and play a role in sustained feeding responses to a cold challenge, even if they are not relevant to the initial hyperphagic response. Thus, future investigation of the role of hormones in sustaining the hyperphagic state during sustained cold exposure is warranted.

Recent work has established that in addition to negative feedback regulation by humoral signals relevant to body fuel stores, AgRP neuron activity is also influenced by 'feed-forward' signals that promote homeostasis by anticipating future changes to energy balance (*Chen and Knight, 2016*; *Lowell, 2019*). Among relevant feed-forward signals are cues from the environment that anticipate eating (e.g., the sight or smell of food), which are presumably communicated indirectly to AgRP neurons via cortical areas that process the relevant sensory input (*Livneh et al., 2017*). How this information is ultimately transmitted to AgRP neurons is unknown, but synaptic relays in the hypothalamic paraventricular nucleus (PVN) (*Krashes et al., 2014*), dorsomedial nucleus (DMH) (*Garfield et al., 2016*), and pre-optic area (POA) may play a role. The key point is that although food intake driven by AgRP neuron activation is commonly associated with states of negative energy balance and/or falling leptin levels, these neurons can also be activated by sensory-related stimuli (*Chen and Knight, 2016*; *Garfield et al., 2016*; *Krashes et al., 2014*).

These considerations have relevance to our finding that the activity of AgRP neurons rapidly increases following exposure to a cold environment, and that the activity of these neurons at thermoneutrality is lower. Additional support for this concept stems from recent evidence that AgRP neuron activity is regulated by sensed temperature in neonatal mice (*Zimmer et al., 2019*). Specifically, exposure of P10 mice to a warm environment rapidly suppresses AgRP neuron activity, whereas pup isolation (with reduced thermal insulation from the nest, dam, and littermates) increases AgRP neuron activity (*Zimmer et al., 2019*). An important distinction from our work, however, is that in neonatal mice, these changes of AgRP neuron activity had no impact on food intake, consistent with previous work suggesting that control of food intake by AgRP neurons does not emerge until weaning (*Gropp et al., 2005*; *Luquet et al., 2005*).

Another key finding reported herein is that in adult mice, the time course of increased AgRP neuron activity coincides with cold-induced increases of both energy expenditure and energy intake, and that each of these responses occur quite rapidly. Thus, by commencing continuous monitoring upon placing mice in a pre-cooled cage (rather than waiting for the cage to gradually cool to the desired temperature), we were able to show that the thermogenic response needed to maintain core body temperature for the duration of the study was achieved within 5 min. Even more impressive is that cold-induced hyperphagia had a similarly rapid onset, at least in mice studied during the mid-light cycle, when they are normally inactive and consume only a small percentage of their daily calories (*Ellacott et al., 2010*). We interpret the rapidity and synchrony with which exposure to a cold environment activates AgRP neurons and engages feeding responses as being consistent with control via a 'feed-forward' mechanism, and future studies are warranted to test this hypothesis.

Our findings further show that chemogenetic inhibition of AgRP neurons selectively blocks cold-induced hyperphagia without impacting cold-induced increases of energy expenditure, RQ or ambulatory activity, suggesting cold-induced AgRP neuron activation is required for the feeding, but not the thermogenic response to this challenge. Initially, these observations seem at odds with published evidence that reduced energy expenditure accompanies hyperphagia induced by chemogenetic activation of AgRP neurons (*Cavalcanti-de-Albuquerque et al., 2019*; *Krashes et al., 2011*), by fasting (*Cahill et al., 1966*), or by ghrelin administration (*Tschöp et al., 2000*). This arrangement makes

sense teleologically given that during fasting, the overarching need of fasted animals is to conserve fuel. In this setting, AgRP neurons are thought to play a pivotal role in both the drive to feed and the suppression of unnecessary fuel utilization, including thermogenesis. Cold exposure, alternatively, reflects a different physiological need state, wherein preservation of core body temperature is paramount, and this cannot be achieved without increased thermogenesis. Given these competing needs, it therefore seems paradoxical that AgRP neuron activation would be an essential component of the adaptive response to both fasting and cold exposure. Adding further complexity is the fact that fasting-associated AgRP neuron activation also inhibits the reproductive axis (*Padilla et al., 2017*), whereas cold-exposed animals continue to reproduce (*Barnett, 1965*).

One explanation for this seeming paradox is that functionally distinct subsets of AgRP neurons exist (*Sternson and Atasoy, 2014*), with those involved in feeding drive in the cold being distinct from those that control energy expenditure or reproductive function. In support of this hypothesis, our Fos staining shows that cold exposure activates only a subset of AgRP neurons, whereas fasting and ghrelin administration each activate the majority of AgRP neurons (*Andrews et al., 2008*; *Liu et al., 2012*). Similarly, in the context that our animals were fed ad libitum for our studies, and AgRP neuron responsiveness is known to be influenced by feeding state (*Chen et al., 2015*), we found that the magnitude of our cold-evoked calcium activity in AgRP neurons is less than what is observed following either fasting or ghrelin administration (*Chen et al., 2015*). These observations collectively support a model in which distinct subpopulations of AgRP neurons, projecting to different downstream brain areas, mediate opposing effects on energy intake and energy expenditure, and that cold exposure activates only the subset that stimulates feeding. Future studies to identify the downstream projection sites and determine the contribution of AgRP, NPY, or the inhibitory neurotransmitter, GABA, to cold-induced feeding responses (*Chen et al., 2019*; *Krashes et al., 2013*) are a priority.

In conclusion, we report that AgRP neuron activation occurs rapidly during cold exposure, and that this response plays a key role to drive the associated hyperphagic, but not the thermogenic, response to this stimulus. Further, we demonstrate that the contribution made by AgRP neuron activation to food consumption is dependent on ambient temperature, constituting a large fraction of intake during cold exposure while playing a lesser role in a thermoneutral environment. In addition to advancing our understanding of the thermoregulatory system, insights into the neurocircuitry linking thermoregulation to AgRP neuron activity may help to identify novel strategies for obesity treatment by blunting the associated hyperphagic response.

# Materials and methods

## Key resources table

| Reagent type (species) or resource | Designation | Source or reference | Identifiers | Additional information |
|---|---|---|---|---|
| Genetic reagent (*Mus musculus*, males) | C57/Bl6/J | Jackson Labs | RRID:IMSR_JAX:000664 | |
| Genetic reagent (*Mus musculus*, males) | B6; Agrptm1 (cre) Lowl/J | Jackson Labs | RRID:IMSR_JAX:012899 | Gift from Steamson Chua, Jr |
| Genetic reagent (*Mus musculus*, males) | Agrp-Cre:GFP | Laboratory of Dr. Richard Palmiter | N/A | |
| Antibody | Anti-Fos (rabbit polyclonal) | Millipore | RRID:AB_2106755 | IF (1:10,000) |
| Antibody | Anti-GFP (Chicken polyclonal) | Abcam ab13970 | RRID:AB_300798 | IF (1:10,000) |

*Continued on next page*

*Continued*

| Reagent type (species) or resource | Designation | Source or reference | Identifiers | Additional information |
|---|---|---|---|---|
| Recombinant DNA reagent | AAV1-CAG-DIO-hM4Di-YFP-WPRE-bGHpA | A gift from Dr. Larry Zweifel (*Sanford et al., 2017*) | NA | |
| Recombinant DNA reagent | AAVDJ-EF1a-DIO-GCaMP6s-WPRE | UNC Viral Core | NA | |
| Recombinant DNA reagent | AAV-Ef1a-DIO-eYFP-WPRE-pA | UNC Viral Core | NA | |
| Recombinant DNA reagent | pAAV-hSyn-DIO-mCherry | Addgene | RRID:Addgene_50459 | |
| Sequence-based reagent | Agrp_Forward primer | Agrp_F | PCR primers | For: 5'-ATGCTGACTCG AATGTTGCTG-3', |
| Sequence-based reagent | Agrp_Reverse primer | Agrp_R | PCR primers | Rev: 5'-CAGACTTAGACC TGGGAACTCT-3' |
| Sequence-based reagent | Pomc_Forward primer | Pomc_F | PCR primers | For: 5'-CAGTGCCA GGACCTCAC-3' |
| Sequence-based reagent | Pomc_Reverse primer | Pomc_R | PCR primers | Rev: 5'-CAGCGAGAGG TCGAGTTTG-3' |
| Sequence-based reagent | Npy_Forward primer | Npy_F | PCR primers | For: 5'-CTCCGCTCTG CGACACTAC-3' |
| Sequence-based reagent | Npy_Reverse primer | Npy_R | PCR primers | Rev: 5'-AGGGTCTTCAA GCCTTGTTCT-3' |
| Sequence-based reagent | Pmch_Forward primer | Pmch_F | PCR primers | For: 5'-GAATTTGGAAGA TGACATAGTAT-3' |
| Sequence-based reagent | Pmch_Reverse primer | Pmch_R | PCR primers | Rev: 5'-CCTGAGCATGTC AAAATCTCTCC-3' |
| Sequence-based reagent | 18S Ribosomal protein_Forward primer | 18S_F | PCR primers | 5'-CGGACAGGATT GACAGATTG-3' |
| Sequence-based reagent | 18S Ribosomal protein_Reverse primer | 18S_R | PCR primers | Rev: 5'-CAAATCGCTCCA CCAACTAA-3' |
| Chemical compound, drug | Clozapine-N-oxide (CNO) | Sigma Aldrich, Inc | Cat #: C0832 | |
| Software, algorithm | Prism 7 and 8 | GraphPad | RRID:SCR_002798 | |
| Software, algorithm | R Software | R Project for Statistical Computing | RRID:SCR_001905 | |
| Software, algorithm | Matlab | Mathworks | RRID:SCR_001622 | Code provided on GitHub as: Cold_Hyperphagia_Requires_AGRP |
| Other | DAPI stain | Invitrogen | Cat #: D1306 | |
| Other | Fluoromount-G | Thermo Fisher Scientific | Cat #: 004958–02 | |

## Animals

All procedures were performed in accordance with NIH Guide for the Care and Use of Laboratory Animals and were approved by the Institutional Animal Care and Use Committee at the University of

Washington. Mice were individually housed in a temperature-controlled room with either a 12:12 hr or 14:10 hr light:dark cycle under specific pathogen-free conditions and were provided with ad libitum access to water and fed a standard laboratory chow (5001; 13% kcal fat, LabDiet, St. Louis, MO). Adult male C57Bl/6 wild-type mice were obtained from Jackson Laboratories, ME, while AgRP-IRES-Cre mice (kindly provided to us by Dr. Streamson Chua, Jr, Albert Einstein College of Medicine) are readily available from Jackson Laboratories, and have been previously described (*Tong et al., 2008*). The AgRP-Cre:GFP knock-in mice (version 2, v2) used in this study were generated by Dr. Richard Palmiter (University of Washington) by replacing the Cre:GFP cassette of the original line (*Sanz et al., 2015*) with a new cassette designed to have attenuated expression of Cre: GFP. The new cassette differs from the original by: (a) removing the nuclear localization signal from Cre, (b) using a non-optimal initiation codon, (c) removing part of the N-terminal sequence of Cre, and (d) adding a 3' untranslated region from the Myc gene that promotes a short mRNA half-life. Crossing mice with this new (v2) line with a conditional reporter line of mice (Gt(ROSA)26-LSL-TdTomato, Allen Institute Seattle WA) gives faithful expression only in AgRP neurons, unlike the original line (*Sanz et al., 2015*) which occasionally resulted in ectopic expression in many parts of the brain. The new targeting construct was electroporated into G4 ES cells (C57Bl/6 × 129/SV) and clones with correct targeting were identified by southern blot of DNA digested with BamH1; 6 of 48 clones analyzed were correctly targeted. Three of these clones were injected into blastocysts from C57Bl/6 mice and one of them gave chimeras with a high percentage of agouti color. Those chimeras were bred with Gt(ROSA)26-FLPer mice to remove the frt-flanked Neo gene. Thereafter, the mice were backcrossed to C57Bl/6 mice for more than six generations before our experiments were performed.

## Viral constructs

Chemogenetic inhibition of AgRP neurons was achieved by microinjection of an AAV containing Cre-dependent cassette for the inhibitory DREADD, AAV1-CAG-DIO-hM4Di-YFP-WPRE-bGHpA (hM4Di: kindly provided by the laboratory of Dr. Larry Zweifel, University of Washington) or mCherry fluorophore control (AAV-hSyn-DIO-mCherry, Addgene, Watertown, MA), into brain areas containing Cre-expressing neurons (e.g., the ARC of AgRP-IRES-Cre mice). Activation of the DREADD receptor was induced by intraperitoneal administration of the agonist, clozapine-N-oxide (CNO, 1 mg/kg, i.p.) (#C0832, Sigma-Aldrich, St. Louis, MO). For fiber-photometry experiments, we utilized an AAV containing a Cre-dependent cassette for the genetically encoded calcium indicator, GCaMP6s, AAVDJ-EF1a-DIO-GCaMP6s-WPRE (UNC Viral Core, Chapel Hill, NC) or eYFP fluorophore control (AAV-Ef1a-DIO-eYFP-WPRE-pA, UNC Viral Core, Chapel Hill, NC).

## Surgery

Stereotaxic viral injections were performed as described (*Faber et al., 2018*; *Meek et al., 2016*). Briefly, animals were anesthetized using 1–3% isoflurane, their head shaved, and placed in three-dimensional stereotaxic frame (Kopf 1900, Cartesian Research Inc, CA). For inhibitory DREADD experiments, the skull was exposed with a small incision, and two small holes were drilled for bilateral microinjection (400 nl per side) of the inhibitory DREADD (hM4Di) AAV into the ARC of AgRP-IRES-Cre mice at stereotaxic coordinates based on the Mouse Brain Atlas: A/P: −1.2, M/L: ±0.3, D/V: −5.85 (*Franklin and Paxinos, 2013*). For fiber photometry experiments, AAV was injected using a unilateral and angled approach (A/P: −1.85, D/V: −6.2, ~12.5° angle from midline). After viral injections, a fiber-optic ferrule (0.48 NA, Ø400 μm core; Doric Lenses, Quebec, Canada) was implanted using the same coordinates. All microinjections were performed using a Hamilton syringe with a 33-gauge needle at a flow rate of 100 nl/min (Micro4 controller, World Precision Instruments, Sarasota, FL), followed by a 5 min pause and slow withdrawal. Animals received a perioperative dose of buprenorphine hydrochloride (0.05 mg/kg sc; Reckitt Benckiser, Richmond, VA). After surgery, mice were allowed 3 weeks to recover to maximize virally transduced gene expression and to acclimate animals to handling and experimental paradigms prior to study. Expression and fiber placement were verified post hoc in all animals, and any data from animals in which the transgene expression and/or fiber was located outside the targeted area were excluded from analysis.

## Body composition analysis

Determination of body fat and lean mass was performed using quantitative magnetic resonance spectroscopy (EchoMRI, Houston, TX) in conscious mice using the NIDDK-funded Nutrition Obesity Research Center Energy Balance Core at the University of Washington (*Taicher et al., 2003*).

## Indirect calorimetry

For indirect calorimetry studies, C57Bl/6 mice were acclimated to metabolic cages after which energy expenditure was measured using a computer-controlled indirect calorimetry system (Promethion, Sable Systems, Las Vegas, NV) as described (*Kaiyala et al., 2015*; *Kaiyala et al., 2012*; *Kaiyala et al., 2016*) with support from the Nutrition Obesity Research Center Energy Balance Core. For each animal, $O_2$ consumption and $CO_2$ production were measured for 1 min at 5 min (acute studies) or 10 min (chronic studies) intervals. RQ was calculated as the ratio of $CO_2$ production to $O_2$ consumption. Energy expenditure was calculated from $VO_2$ and $VCO_2$ data using the Weir equation (*Weir, 1949*). Ambulatory activity was measured continuously with consecutive adjacent infrared beam breaks in the x-, y-, and z-axes were scored as an activity count that was recorded every 5 or 10 min. Data acquisition and instrument control were coordinated by MetaScreen v.1.6.2 and raw data was processed using ExpeData v.1.4.3 (Sable Systems) using an analysis script documenting all aspects of data transformation.

## Core body temperature monitoring

Adult male C57Bl/6 mice received body temperature transponders implanted into the peritoneal cavity (Starr Life Science Corp, Oakmont, PA) and were allowed a 1-week recovery period. Animals were then acclimated to metabolic cages enclosed in temperature- and humidity-controlled cabinets (Caron Products and Services, Marietta, OH) prior to study. Signals emitted by body-temperature transponders were sensed by a platform receiver positioned underneath the cage and analyzed using VitalView software as described (*Kaiyala et al., 2015*; *Kaiyala et al., 2016*).

## Immunohistochemistry

For immunohistochemical studies, animals were overdosed with ketamine:xylazine and perfused with $1\times$ phosphate-buffered saline (PBS) followed by 4% (v/v) paraformaldehyde in 0.1M PBS. Brains were removed and post-fixed for 4 hr in paraformaldehyde followed by sucrose (30%) dehydration and embedding in OCT blocks. Free-floating coronal sections were obtained via Cryostat at 35 μm thickness and stored in $1\times$ PBS with 0.02% sodium azide at 4°C. Free-floating sections were then washed at room temperature in phosphate-buffered saline with 0.1% Tween 20 or 0.4% Triton-X 100 (PBS-T) for 1 hr, followed by a blocking buffer (5% normal donkey serum, 1% bovine serum albumin in 0.1M PBS-T with 0.01% sodium azide) for an additional hour with rocking. Sections were then incubated 24–48 hr at 4°C with polyclonal rabbit anti-Fos (Ab-5, 4–17), 1:10,000; (Millipore, Burlington, MA) and/or Chicken anti-GFP (Abcam, ab13970, 1:10,000; Cambridge, UK) in blocking buffer, followed by PBS-T washes at room temperature. Sections were then incubated in secondary donkey anti-rabbit Alexa 594 or donkey anti-chicken Alexa 488 (1:500, Jackson ImmunoResearch Laboratories, West Grove, PA) in blocking buffer overnight at 4°C, followed by PBS-T washes. Sections were stained with DAPI (1:10,000, Sigma, St. Louis, MO) for 30 min, followed by a final set of washes and mounting with Fluoromount-G (Thermo Fisher Scientific, Wilmington, DE) or prepared polyvinyl acetate (PVA).

## qRT-PCR

To quantify specific hypothalamic mRNA transcripts, mice were sacrificed at study completion and the hypothalamus rapidly dissected and flash frozen. Briefly, the brain was removed and, using a brain matrix, a 2.0 mm thick coronal slice including the entire hypothalamus was made. Then, a Harris Unicore 1.0 mm tissue punch (Ted Pella, Inc, Redding, CA) was used to dissect the hypothalamus such that the total tissue punch was a cylinder 2.0 mm in length and 1.0 mm in diameter, providing consistency in the amount of RNA isolated. Individual tissue samples were homogenized (Dounce) and RNA was isolated using Qiagen RNeasy Micro Kit (Kit# 74004, Hilden, Germany) and isolated RNA concentrations were quantified by Nanodrop (Thermo Fisher Scientific, Wilmington, DE). qRT-PCR was performed using SYBR Green One-Step (Kit# 600825, Agilent, Santa Clara, CA). qRT-PCR

data were analyzed using the Sequence Detection System software (SDS Version 2.2; Applied Bio-systems, Foster City, CA). Expression levels of each gene were normalized to a housekeeping gene (18S RNA) and standard curve. Non-template controls were incorporated into each PCR run. Selected oligonucleotides are found in the Key Resources Table.

## Fiber photometry

Mice expressing GCaMP6s in AgRP neurons were connected to a fiber-photometry system to enable fluorometric analysis of real-time neuronal activity. Briefly, for calcium recording in vivo, two excitation wavelengths (470 nm and 405 nm isosbestic) were used to indicate calcium-dependent and calcium-independent (i.e., due to bleaching and motion artifacts) GCaMP6s fluorescence, respectively. Light was delivered via fiber-coupled LEDs (LED lights: M470F3 and M405FP1, LED driver: DC4104, Thorlabs, Newton, NJ) and modulated by a real-time amplifier (RZ5P, Tucker-Davis Technology (TDT), Alachua, FL) at non-divisible frequencies (331 Hz and 231 Hz, respectively) to prevent signal interference between the channels. Excitation lights were bandpass filtered (475 ± 15 nm, 405 ± 5 nm; iFMC4, Doric Lenses, Quebec, QC, Canada) and the combined excitation light delivered through a fiber-optic patch cord (M75L01,Thorlabs) connected to a rotary joint (FRJ, 0.48 NA, Ø400 µm core; Doric Lenses) to prevent fiber-optic torsion during animal movement. A final connector patch cord (MFP, 0.48 NA, Ø400 µm core; Doric Lenses) was connected to the implanted fiber-optic via a ceramic mating sleeve (ADAL1, Thorlabs). Emitted light was collected through the same patch cord, bandpass filtered (525 ± 25 nm; iFMC4) and transduced to digital signals by an integrated photodetector head. Electrical signals were sampled at a rate of 1017.25 Hz and demodulated by the RZ5P real-time processor. Experiments were controlled by Synapse software (TDT).

Custom MATLAB scripts were developed for analyzing fiber photometry data. The isosbestic 405 nm excitation control signal was subtracted from the 470 nm excitation signal to remove movement artifacts from intracellular calcium-dependent GCaMP6s fluorescence. Baseline drift was evident in the signal due to slow photobleaching artifacts, particularly during the first several minutes of each recording session. A double exponential curve was fit to the raw trace of temperature-ramping experiments while a linear fit was applied to the raw trace of food presentation experiments and subtracted to correct for baseline drift. After baseline correction, dF/F (%) was calculated as individual fluorescence intensity measurements relative to median fluorescence of entire session for 470 nm channel. Averaged dF/F (%) for each temperature (30 or 14°C) were limited to the 10 min period either before or after the 1 min ramp either between the 14°C and 30°C transition or the 30°C and 14°C transition.

## Measuring determinants of energy balance

To examine the effect of chronic mild, cold exposure on comprehensive measures of energy homeostasis, mice were acclimated to metabolic cages housed within temperature- and humidity-controlled chambers, and either remained at 22°C or they were exposed to 14°C for 5 days. During this period, continuous measures of energy expenditure, RQ, ambulatory activity, and energy and water intake were recorded, and body composition was determined both before and after the thermogenic challenge.

To examine the effect of acute changes in ambient temperature on energy intake, wild-type mice (mean BW: 24.59 ± 0.31 g) were acclimated to metabolic cages housed within temperature- and humidity-controlled chambers. One-hour fasted mice were placed directly into metabolic cages housed within environmental chambers that were pre-set overnight at either 22°C or 14°C at 10 AM for 24 hr for continuous measures of energy expenditure, RQ, ambulatory activity, and energy intake. Animals were studied using a randomized crossover design in which housing conditions were separated by at least 72 hr, with each animal serving as its own control.

## Measuring Fos induction

To determine whether acute cold exposure induces Fos in AgRP neurons, sated AgRP-Cre:GFP mice were housed in temperature-controlled chambers pre-set overnight at either 22°C, 30°C, or 14°C for 90 min. Although food is typically removed during the period prior to Fos quantitation in this type of study (so as to minimize the impact that variation in food intake can have on the activity of AgRP neurons), we opted not to do so here, based on the concern that cold-exposed mice would

experience a state of negative energy balance greater than occurred in controls housed at room temperature (owing to a comparatively greater rate of energy expenditure), and that this might confound data interpretation. Animals were then anesthetized and perfused as described below. Immunostaining for Fos was quantified by imaging at either 10× or 20× on a Leica SP8X confocal system with support from the University of Washington, W.M. Keck Microscopy Center. Images were merged using ImageJ (Fiji, NIH) and threshold adjusted to minimize nonspecific background fluorescence. Fos+ cells were then identified and counted in 35 μm sections obtained serially across the full rostral to caudal axis of the arcuate nucleus using the 'analyze particles' feature, such that consistent fluorescence and size thresholds were used throughout, as previously described (*Faber et al., 2018*). For measures of Fos induction in additional thermoregulatory nuclei, two to three sections from each region were counted for each subject.

## Measuring AgRP neuron activity

To assess the impact of cold exposure on AgRP neuron activity, mice were acutely housed on a custom thermal platform. Baseline GCaMP6s fluorescence signals were set to similar levels across animals by adjusting the intensities of the 470 nm and 405 nm LEDs, and a baseline recording was measured for 5 min. For temperature-challenge studies, mice were placed in a small custom-built plexiglass temperature chamber (3' × 6' × 6') that was constructed to enclose a Peltier cooler platform that could be controlled by an external controller (TE Tech, TC720) as described (*Tan et al., 2016*). Animals were acclimated on three separate days to tether and Peltier platform. For temperature ramp studies, animals were attached to tether and allowed 2–3 min for photobleaching before photometry recording is initiated. Temperature-ramp studies were designed to test GCaMP6s activity in AgRP neurons when transitioning from 30℃ to 14℃ or 14℃ to 30℃. Animals were held at 30℃ for 10 min before transitioning to 14℃ for 10 min before returning to 30℃ with each temperature ramp repeated twice during a recording session. Transitions between temperatures were all set to 60 s. Experiments were aligned to initial 22℃. As an added positive control, at the end of each study, animals were presented with a food pellet, which led to an expected reduction in AgRP neuron activity.

Prior to study, mice were acclimated to experimental procedures and AgRP-calcium responses to food presentation in a fast-refeed paradigm was used as a positive control to indicate successfully targeted animals. Briefly, overnight-fasted animals that failed to demonstrate ≥10% reduction in F/F in response to chow presentation were identified as surgical misses and excluded from further study; post hoc IHC validated poor viral expression and/or fiber-optic placement in these animals.

## DREADD-mediated inhibition of AgRP neurons

To determine whether activation of AgRP neurons is required for cold-induced hyperphagia, AgRP-IRES-Cre mice (mean BW: 36.10 ± 1.06 g) received a bilateral microinjection of the inhibitory DREADD (hM4Di) or mCherry fluorophore control directed to the ARC. After 3 weeks for recovery from surgery, acclimation to injection/handling, and expression of the transgene, animals were acclimated to environmental chambers to minimize stress. On the day prior to the experiment, animals were returned to their previous home cage and the environmental chambers were allowed 16 hr to come to temperature (30℃, 22℃, or at 14℃). On the experimental day, food was removed at 9 AM and mice received a pretreatment injection of either CNO (1 mg/kg, i.p.) or saline in a randomized crossover manner. At 10 AM, mice were placed into metabolic cages housed within environmental chambers set at either 30℃, 22℃, or 14℃ with continuous measures of energy intake and energy expenditure recorded over a 4 hr period.

## Statistics

Results are expressed as mean ± SEM. Significance was established at p<0.05, two-tailed. For statistical comparisons involving core temperature, energy expenditure, ambulatory activity, RQ or energy intake, data obtained during chronic cold exposure (14℃) relative to room temperature (22℃) were reduced into average light and dark photoperiods for each mouse. A group by ambient temperature ANOVA with least significance difference pairwise tests compared mean values between groups. Where applicable, time-series data were analyzed using treatment by time-mixed factorial ANOVA or linear-mixed model analysis controlling for within-subjects correlated data using a random

intercept model with treatment and time as fixed effects (*Fitzmaurice et al., 2012*). In studies with multiple comparisons made between two independent groups, two-way ANOVA was applied with Holm-Sidak correction for multiple comparisons, as requested. All post hoc comparisons following one-way ANOVA were determined using Sidak correction for multiple comparisons. A two-sample unpaired Student's t-test was used for two-group comparisons and a paired t-test for within group comparisons. Statistical analyses were performed using Statistica (v7.1; Statsoft, Incl, Tulsa, OK), SPSS (SPSS version 23, IBM Corp., Somers, NY), R (v 3.6.2, R Project for Statistical Computing, RRID: SCR_001905), MATLAB (Mathworks; Natick, MA) (https://github.com/christianepedersen/Cold_Hyperphagia_Requires_AGRP; copy archived at swh:1:rev:6af9ba4a172f2ee08454f906382b-fe29b7c8fdbf [*Deem, 2021* ] and GraphPad Prism [version 7.0; La Jolla, CA]).

## Acknowledgements

The authors gratefully acknowledge Dr. Larry Zweifel (University of Washington) for providing the inhibitory DREADD virus. This work was supported by National Institutes of Health Grants DK089056 and DK124238 (to GJM), DK083042 and DK101997 (to MWS), R37DA033396 (to MRB), R01-DA-24908 (to RDP), the NAPE Center Imaging and Circuits Core (P30 DA048736), the University of Washington WM Keck Microscopy Center (S10 OD016240), the Nutrition Obesity Research Center (DK035816), the Diabetes Research Center (DK17047), F31 DK113673 and T32 GM095421 (to CLF), and the Nutrition, Obesity and Atherosclerosis Training Grant (T32 HL007028) at the University of Washington (to JDD), a Dick and Julia McAbee Endowed Fellowship (to JDD), and an American Diabetes Association Innovative Basic Science Award (ADA 1–19-IBS-192, to GJM) and Fellowship Grant (ADA 1–19-PDF-103, to JDD).

## Additional information

### Funding

| Funder | Grant reference number | Author |
|---|---|---|
| National Institutes of Health | DK089056 | Gregory J Morton |
| National Institutes of Health | DK083042 | Michael W Schwartz |
| National Institutes of Health | DK101997 | Michael W Schwartz |
| National Institutes of Health | R37 DA033396 | Michael R Bruchas |
| National Institutes of Health | R01DA24908 | Richard D Palmiter |
| National Institutes of Health | P30 DA048736 | Michael R Bruchas |
| National Institutes of Health | DK035816 | Gregory J Morton |
| Diabetes Research Center | DK17047 | Gregory J Morton |
| National Institutes of Health | F31 DK113673 | Chelsea L Faber |
| National Institutes of Health | T32 GM095421 | Chelsea L Faber |
| National Institutes of Health | T32 HL007028 | Jennifer D Deem |
| Diabetes Research Center | | Jennifer D Deem |
| American Diabetes Association | ADA 1-19-PDF-103 | Jennifer D Deem |
| University of Washington | S10 OD016240 | Michael R Bruchas |
| American Diabetes Association | ADA 1–19-IBS-192 | Gregory J Morton |
| University of Washington | Dick and Julia McAbee Endowed Fellowship | Jennifer D Deem |
| National Institutes of Health | F31 DA051124 | Christian Pedersen |

The funders had no role in study design, data collection and interpretation, or the decision to submit the work for publication.

## Author contributions

Jennifer D Deem, Conceptualization, Data curation, Formal analysis, Supervision, Funding acquisition, Validation, Investigation, Visualization, Methodology, Writing - original draft, Project administration, Writing - review and editing; Chelsea L Faber, Richard D Palmiter, Methodology, Writing - review and editing; Christian Pedersen, Formal analysis, Methodology; Bao Anh Phan, Data curation, Methodology; Sarah A Larsen, Kayoko Ogimoto, Methodology; Jarrell T Nelson, Megan A Tran, Data curation; Vincent Damian, Validation; Karl J Kaiyala, Formal analysis; Jarrad M Scarlett, Michael W Schwartz, Conceptualization, Funding acquisition, Writing - review and editing; Michael R Bruchas, Funding acquisition, Writing - review and editing; Gregory J Morton, Conceptualization, Data curation, Formal analysis, Funding acquisition, Methodology, Project administration, Writing - review and editing

## Author ORCIDs

Jennifer D Deem (iD) https://orcid.org/0000-0002-8865-5145
Chelsea L Faber (iD) http://orcid.org/0000-0002-4812-8164
Richard D Palmiter (iD) http://orcid.org/0000-0001-6587-0582
Michael R Bruchas (iD) http://orcid.org/0000-0003-4713-7816
Michael W Schwartz (iD) http://orcid.org/0000-0003-1619-0331
Gregory J Morton (iD) https://orcid.org/0000-0002-8106-8386

## Ethics

Animal experimentation: This study was performed in strict accordance with the recommendations in the Guide for the Care and Use of Laboratory Animals of the National Institutes of Health. All of the animals were handled according to a protocol approved by the institutional animal care and use committee (IACUC) of the University of Washington (#2456-06). All surgery was performed under isoflurane anesthesia, and every effort was made to minimize suffering.

## Decision letter and Author response

Decision letter https://doi.org/10.7554/eLife.58764.sa1
Author response https://doi.org/10.7554/eLife.58764.sa2

# Additional files

## Supplementary files

• Transparent reporting form

## Data availability

Photometry data has been deposited in Dryad https://doi.org/10.5061/dryad.0p2ngf208. Individual source data files are associated with individual figures.

The following dataset was generated:

| Author(s) | Year | Dataset title | Dataset URL | Database and Identifier |
|---|---|---|---|---|
| Deem JD | 2020 | Cold-induced hyperphagia requires AgRP-neuron activation in mice | https://doi.org/10.5061/dryad.0p2ngf208 | Dryad Digital Repository, 10.5061/dryad.0p2ngf208 |

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
