## [Decision Letter]

**Acceptance summary:**

This study investigates the neural basis of cold-induced hyperphagia. AGRP neurons are engaged quickly during cold exposure- as indicated by increased AGRP/NPY expression, increased cFos levels, and fiber photometry recordings. Cold-induced hyperphagia is apparently not secondary to increased energy expenditure (as one might have expected) but instead seems to involve a neural pathway from temperature-sensing neurons to AGRP neurons.

**Decision letter after peer review:**

Thank you for submitting your article "Cold-induced hyperphagia requires AgRP neuron activation in mice" for consideration by *eLife*. Your article has been reviewed by three peer reviewers, including Stephen Liberles as the Reviewing Editor and Reviewer #1, and the evaluation has been overseen by Gary Westbrook as the Senior Editor. The following individual involved in review of your submission has agreed to reveal their identity: Matthew Carter (Reviewer #2). The reviewers have discussed the reviews with one another and the Reviewing Editor has drafted this decision to help you prepare a revised submission.

We would like to draw your attention to changes in our revision policy that we have made in response to COVID-19 (https://elifesciences.org/articles/57162). Specifically, when editors judge that a submitted work as a whole belongs in *eLife* but that some conclusions require a modest amount of additional new data, as they do with your paper, we are asking that the manuscript be revised to either limit claims to those supported by data in hand, or to explicitly state that the relevant conclusions require additional supporting data. Our expectation is that the authors will eventually carry out the additional experiments and report on how they affect the relevant conclusions either in a preprint on bioRxiv or medRxiv, or if appropriate, as a Research Advance in *eLife*, either of which would be linked to the original paper.

Summary:

This study investigates the neural basis of cold-induced hyperphagia. AGRP neurons are engaged quickly during cold exposure- as indicated by increased AGRP/NPY expression, increased cFos levels, and fiber photometry recordings. Cold-induced hyperphagia is apparently not secondary to increased energy expenditure (as one might have expected) but instead seems to involve a neural pathway from temperature-sensing neurons to AGRP neurons.

Essential revisions for this paper:

All reviewers found the data to be generally convincing, but requested some additional clarifications. Of particular note were controls in DREADD experiments (reviewer #2, point #2) and controls in fiber photometry experiments to judge the magnitude of responses (reviewer #3, point #1). If data is not available, the reviewers suggested discussing these concerns, and for the fiber photometry experiments, to at least present the data in an absolute ΔF/F scale for easier comparison to published work.

Revisions expected in follow-up work:

Please see other comments below for textual revisions requested and other experiments expected in follow-up work.

Reviewer #1:

This is a nice and important study, and I support its publication in *eLife*. The authors paint a convincing picture that AGRP neurons are engaged quickly during cold exposure- as indicated by increased AGRP/NPY expression, increased cFos levels, and fiber photometry recordings. Cold-induced hyperphagia is apparently not secondary to increased energy expenditure (as one might have expected) but instead seems to involve a neural pathway from temperature-sensing neurons to AGRP neurons, which is surprising and interesting. I do note that the DREADD experiments are a bit of a sledgehammer- any conditions that promote feeding would presumably be impacted by this manipulation, and there does seem to be a trend in 5F which might be significant with greater n, or slightly different experimental conditions (such as using food-restricted mice). The ideal experiment would be to block only the specific AGRP neuron inputs that mediate cold responses but this could be the subject of future exciting work. With the manuscript in its current form, the overall model is convincing, findings represent an important advance, and I support publication as is. My comments are below, and given the ongoing pandemic, do not request any additional experiments. If such data is already available, it would be nice to include, but is not a requirement for publication.

1) To supplement Figure 1H, is RNA in situ hybridization data already available for *Agrp*/*Npy*/*Pomc* transcripts at 22/14 degrees?

2) To supplement Figure 3, is control data quantifying Fos+/POMC+ cells already available?

3) Is fiber photometry data already available about the effect of cold on arcuate POMC neurons?

4) The Dietrich Lab recently published a role for AGRP neurons in pup hypothermia (Zimmer et al., 2019) which should be briefly discussed.

Reviewer #2:

The manuscript by Deem et al. explores the necessity of AgRP neuronal activity in the phenomenon of cold-induced hyperphagia in mice. The authors present evidence that AgRP neurons are activated rapidly upon exposure to cold temperatures using Fos staining and fiber photometry. The authors then use chemogenetics to demonstrate that bilateral inhibition of AgRP neurons blocks the increase in food intake observed upon exposure to colder temperatures (but does not block an increase in thermogenesis).

Taken together, these experiments seem well-executed and the results justify the Title and main conclusions of the study, that cold-induced hyperphagia requires AgRP neuron activation in mice. There are only a small number of substantive concerns:

1) A few comments about statistics and data analysis:

a) In Figure 1, why are the gene expression results for *Agrp*, *Npy*, *Pomc*, and *Pmch* plotted in different charts if they are representative of similar datasets? Were they measured from different animals? If not, they could be presented on the same axes. These data would also seem to necessitate a 2-way ANOVA to account for multiple comparisons of the data, rather than a series of student's t-tests.

b) The figure legend for Figure 2 states that a RM 2-way ANOVA was used, but I don't see any visual representation of statistics or reporting of statistics in the test that seems to require a RM 2-way ANOVA?

c) After reporting that a one-way ANOVA was used in Figure 3 and Figure 3—figure supplement 1, it would be helpful to also report the post-hoc test used for multiple comparisons analysis.

d) In the bar chart in Figure 5F, there is an individual dot that doesn't have a connecting line. If these are paired data, then there shouldn't be an individual in the CNO group that is not also part of the saline group?

e) In Figures 2—figure supplement 1 and Figure 5—figure supplement 3, the term "ns" is used to imply "not significant," but no statistical tests are reported in the legends.

2) The control groups for the chemogenetic experiments (Figure 5) use an injection of saline instead of an injection of CNO into animals transduced with hM4Di. The gold standard control group for a chemogenetic experiment is to inject CNO into animals injected with a non-DREADD transgene (e.g. a fluorescent reporter gene). This control ensures that it is the inhibition of the neurons of interest that mediate the reported effects, rather than CNO itself. Indeed, recent studies have shown that CNO itself does cause off-target effects and is capable of being metabolized by the body [see Gomez et al. (2017) Science; Mahler et al. (2018) Neuropsychopharmacology; Manvich et al. (2018) Scientific Reports]. Therefore, in the key results of Figure 5, the authors cannot state, definitively, that it is inhibition of AgRP neurons that blocks an increase in feeding, as it is possible that the doses of CNO administered (1.0 mg/kg as opposed to the 0.3 mg/kg used in many studies) may block increases in feeding. This point does not necessitate repeating all of the previously performed experiments, but should at least be acknowledged.

Reviewer #3:

This is a well written and thoughtful manuscript providing evidence for the rapid regulation of the response to cold – one that raises a critical question about the role of compensatory feeding. The authors have done a good job of supporting the claim that the food intake, driven likely by activating AgRP neuron circuits, occurs before a negative energy balance has occurred. I am supportive of this manuscript but have a one main data request that is important and a couple of other questions regarding the context of the paper.

Main experimental concerns:

1) All the data are presented as Z scores. This prevents the reader from trying to understand the magnitude of the activity change. For example, in comparison to a ghrelin injection, how much of an increase in AgRP neuron activity is observed? Or could some other metric be used to compare this to other responses seen in the literature?

This may speak to the heterogeneous AgRP neuron activity response (see below) – as only a subset of neurons are activated. Or it could indicate that the activation is less intense then ghrelin. Either way, letting the reader know this comparison would be a valuable addition to the manuscript that would help frame future studies.

2) How is the AgRP neuron inhibition quantified? Is the Fos induced by cold completely gone?

a) This may also speak to why heat production is still intact – what if only half of the AgRP neuron activity was blocked?

Other comments:

3) A paradox is presented here in regards to published literature. Krashes et al., 2011 paper shows AgRP neuron activation decreases energy expenditure. Yet, these neurons are activated in the cold as an animal increase its energy expenditure. Discuss of this is warranted.

4) "Instead, the data are more consistent with a model in which AgRP neuron activation contributes to cold-induced hyperphagia."

I understand that *Agrp* and *Npy* expression are upregulated in hunger, but is this tied uniquely to AgRP neuron activation? Has that experiment been done to enhance this hypothesis?

5) Figure 3A-C is never called out in text.

6) It is not clear from text/figure that 3F means that ~25% of AGRP neurons have Fos after cold exposure. Is that true? And what does that mean? Especially in light of the photometry data.

7) What do the authors make of the large number of non AgRP neurons in the arcuate that express Fos? (Figure 3—figure supplement 2).

8) What is the grey shaded area in Figure 4E?

---

## [Author Response]

Reviewer #1:This is a nice and important study, and I support its publication in eLife. The authors paint a convincing picture that AGRP neurons are engaged quickly during cold exposure- as indicated by increased AGRP/NPY expression, increased cFos levels, and fiber photometry recordings. Cold-induced hyperphagia is apparently not secondary to increased energy expenditure (as one might have expected) but instead seems to involve a neural pathway from temperature-sensing neurons to AGRP neurons, which is surprising and interesting. I do note that the DREADD experiments are a bit of a sledgehammer- any conditions that promote feeding would presumably be impacted by this manipulation, and there does seem to be a trend in 5F which might be significant with greater n, or slightly different experimental conditions (such as using food-restricted mice). The ideal experiment would be to block only the specific AGRP neuron inputs that mediate cold responses but this could be the subject of future exciting work. With the manuscript in its current form, the overall model is convincing, findings represent an important advance, and I support publication as is. My comments are below, and given the ongoing pandemic, do not request any additional experiments. If such data is already available, it would be nice to include, but is not a requirement for publication.1) To supplement Figure 1H, is RNA in situ hybridization data already available for Agrp/Npy/Pomc transcripts at 22/14 degrees?

For Figure 1, hypothalami were dissected for qRT-PCR analysis in animals that were housed at either 22ºC or 14ºC in order to measure multiple hypothalamic neuropeptide transcripts. Consequently, brains are not readily available for quantification of *Agrp/Npy/Pomc* transcripts at 22/14 degrees using in situ hybridization. We do not expect an additional in situ hybridization study to substantially alter our conclusions based on qRT-PCR.

2) To supplement Figure 3, is control data quantifying Fos+/POMC+ cells already available?

Please see the response to point #3 below.

3) Is fiber photometry data already available about the effect of cold on arcuate POMC neurons?

In response to points #2 and #3 above, we agree that investigation into the response of *Pomc* neurons to cold exposure is of strong interest. However, we feel that the information in the current manuscript is of sufficient scientific interest to warrant publication without including investigation into other neuronal cell types. We add that to do so would delay publication of our work by many months and is unlikely to challenge the interpretation of our core observations. We therefore feel that this work is best saved for a future paper.

4) The Dietrich Lab recently published a role for AGRP neurons in pup hypothermia (Zimmer et al., 2019) which should be briefly discussed.

We agree and have expanded our discussion of the work by Zimmer, et al. suggesting neonatal AgRP neuron activity is increased by a cool ambient temperature. Specifically, exposure of P10 mice to a warm environment rapidly suppresses AgRP neuron activity, whereas pup isolation (with reduced thermal insulation from the nest, dam, and littermates) has the opposite effect. With this said, however, an important distinction is that unlike in adult mice, Zimmer et al. report that “neonatal AgRP neuron activity does not stimulate food intake, suggesting that the role of AgRP neurons in feeding drive develops later in development.” This conclusion is consistent with other work suggesting that an effect of AgRP neuron activation to induce feeding behavior is not evident until after weaning (Gropp, et al., 2005, Luquet, et al., 2005).

Revised Discussion text: “These considerations have relevance to our finding that the activity of AgRP neurons rapidly increases following exposure to a cold environment, and that the activity of these neurons at thermoneutrality is lower. […] An important distinction from our work, however, is that in neonatal mice, these changes of AgRP-neuron activity had no impact on food intake, consistent with previous work suggesting that control of food intake by AgRP neurons does not emerge until weaning (Gropp et al., 2005; Luquet et al., 2005).”

Reviewer #2:The manuscript by Deem et al. explores the necessity of AgRP neuronal activity in the phenomenon of cold-induced hyperphagia in mice. The authors present evidence that AgRP neurons are activated rapidly upon exposure to cold temperatures using Fos staining and fiber photometry. The authors then use chemogenetics to demonstrate that bilateral inhibition of AgRP neurons blocks the increase in food intake observed upon exposure to colder temperatures (but does not block an increase in thermogenesis).Taken together, these experiments seem well-executed and the results justify the Title and main conclusions of the study, that cold-induced hyperphagia requires AgRP neuron activation in mice. There are only a small number of substantive concerns:1) A few comments about statistics and data analysis:a) In Figure 1, why are the gene expression results for Agrp, Npy, Pomc, and Pmch plotted in different charts if they are representative of similar datasets? Were they measured from different animals? If not, they could be presented on the same axes. These data would also seem to necessitate a 2-way ANOVA to account for multiple comparisons of the data, rather than a series of student's t-tests.

The *Agrp, Npy, Pomc* and *Pmch* gene expression results are from the same data set. In response to the reviewer’s request, the data are now presented visually such that all genes are included in the same graph, and the legend and text were revised accordingly. In addition, as suggested, we have reanalyzed the data using 2-way ANOVA with Holm-Sidak correction for multiple comparisons, which did not change the conclusions drawn.

Revised Materials and methods subsection “Statistics” text: “In studies with multiple comparisons made between two independent groups, 2-way ANOVA was applied with Holm-Sidak correction for multiple comparisons, as requested.”

b) The figure legend for Figure 2 states that a RM 2-way ANOVA was used, but I don't see any visual representation of statistics or reporting of statistics in the test that seems to require a RM 2-way ANOVA?

The analysis performed on the time series data was a mixed-ANOVA for within-subjects comparison rather than a repeated measures-ANOVA. We have revised the figure to include markers of significance on the time-series data and edited the legend and the Materials and methods section for clarity.

Revised Figure 2 legend text: “(A) Adult male wild-type mice were acutely housed in temperature-controlled chambers set to either mild cold (14°C) or, as a control, room temperature (22°C). […] ****p<0.0001, *p<0.05, based on mixed factorial-ANOVA for changes of food intake over time, and paired Student’s t-test to compare mean values of food intake and heat production.”

c) After reporting that a one-way ANOVA was used in Figure 3 and Figure 3—figure supplement 1, it would be helpful to also report the post-hoc test used for multiple comparisons analysis.

To correction for multiple comparisons when using one-way ANOVA, we used the Sidak method. The figure legends for both Figure 3 and Figure 3—figure supplement 1 have been edited to clarify this.

d) In the bar chart in Figure 5F, there is an individual dot that doesn't have a connecting line. If these are paired data, then there shouldn't be an individual in the CNO group that is not also part of the saline group?

Thank you for pointing this out. The individual dot was included in error, and we take full responsibility for this mistake. The figure has been revised to correct this problem and data reanalyzed to make sure that it is correct. Our overall interpretation of the data has not changed.

e) In Figure 2—figure supplement 1 and Figure 5—figure supplement 3, the term "ns" is used to imply "not significant," but no statistical tests are reported in the legends.

A paired student’s t-test was used to compare the average values of RQ and locomotor activity in these supplementary figures. Text clarifying this analysis has been added to the corresponding figure legends.

2) The control groups for the chemogenetic experiments (Figure 5) use an injection of saline instead of an injection of CNO into animals transduced with hM4Di. The gold standard control group for a chemogenetic experiment is to inject CNO into animals injected with a non-DREADD transgene (e.g. a fluorescent reporter gene). This control ensures that it is the inhibition of the neurons of interest that mediate the reported effects, rather than CNO itself. Indeed, recent studies have shown that CNO itself does cause off-target effects and is capable of being metabolized by the body [see Gomez et al. (2017) Science; Mahler et al. (2018) Neuropsychopharmacology; Manvich et al. (2018) Scientific Reports]. Therefore, in the key results of Figure 5, the authors cannot state, definitively, that it is inhibition of AgRP neurons that blocks an increase in feeding, as it is possible that the doses of CNO administered (1.0 mg/kg as opposed to the 0.3 mg/kg used in many studies) may block increases in feeding. This point does not necessitate repeating all of the previously performed experiments, but should at least be acknowledged.

We thank the reviewer for requesting the inclusion of CNO controls for our feeding studies. In response to the reviewer’s request, we performed an additional study in a new cohort of AgRPIRES-Cre mice that received a bilateral injection of a DIO-eYFP control virus directed to the ARC. Mice then received an ip injection of either CNO or its vehicle (saline) in a randomized manner one hour prior to being placed in metabolic cages housed within temperature-controlled chambers previously set to either 14, 22 or 30°C for continuous measures of energy expenditure and energy intake. As expected, following saline administration, both energy expenditure and energy intake varied dose-dependently with ambient temperature, such that both were highest at the coldest temperature (i.e., 14°C), and the same response was observed in mice receiving CNO (Figure 5—figure supplement 2). By excluding an independent effect of CNO on either energy intake or energy expenditure, this finding further strengthens our conclusion that AgRP neuron activation is required for intact feeding responses to cold exposure.

Revised Results text: “Consistent with this prediction, we found that whereas inhibition of AgRP neurons had little effect on food intake in mice housed at 30°C (Figure 5D), intake was strongly reduced in animals housed at 14°C, while intake was only modestly inhibited by CNO in mice housed at 22°C (Figure 5F and H). […] Moreover, these responses cannot be explained by a nonspecific effect of CNO, as cold-induced increases of food intake and EE in AgRP-IRES-Cre mice expressing a control mCherry fluorophore did not differ following ip injection of CNO vs. saline (Figure 5—figure supplement 2).”

Reviewer #3:This is a well written and thoughtful manuscript providing evidence for the rapid regulation of the response to cold – one that raises a critical question about the role of compensatory feeding. The authors have done a good job of supporting the claim that the food intake, driven likely by activating AgRP neuron circuits, occurs before a negative energy balance has occurred. I am supportive of this manuscript but have a one main data request that is important and a couple of other questions regarding the context of the paper.Main experimental concerns:1) All the data are presented as Z scores. This prevents the reader from trying to understand the magnitude of the activity change. For example, in comparison to a ghrelin injection, how much of an increase in AgRP neuron activity is observed? Or could some other metric be used to compare this to other responses seen in the literature?This may speak to the heterogeneous AgRP neuron activity response (see below) – as only a subset of neurons are activated. Or it could indicate that the activation is less intense then ghrelin. Either way, letting the reader know this comparison would be a valuable addition to the manuscript that would help frame future studies.

While presentation of z-score-normalized fiber photometry data is common practice in other neuroscience fields, we understand it is relatively uncommon in the AgRP neuron field. This is because fiber photometry studies on this neuronal population are typically performed during conditions where the animal is either fasted or calorically restricted, which maximally activates the population resulting in robust and largely uniform signal responses. Indeed, as alluded to by the reviewer, the relative magnitude of the AgRP neuron activity changes in response to ambient temperature (Figure 4) is more subtle relative to fasting or ghrelin administration. This assertion is based on our Fos immunohistochemical data, which shows that only a small subset of AgRP neurons (~25%; Figure 3F) are activated by cold exposure, whereas fasting and peripheral ghrelin administration result in more widespread induction of Fos (Liu et al., 2012, Andrews et al., 2008). This observation, as well reviewer points #3 and #6 below, bears on how best to interpret our findings in the context of relevant literature, and we have taken two key steps in response to these comments. First, we have increased the number of animals in our fiber photometry study, and second, we now present the fiber photometry data as absolute magnitude (dF/F (%)). We are pleased to report that these steps have served to strengthen our interpretation of the data. In addition, we have made revisions in the Materials and methods and Discussion sections.

Revised Discussion text: “In support of this hypothesis, our Fos staining shows that cold exposure activates only a subset of AgRP neurons, whereas fasting and ghrelin administration each activate the majority of AgRP neurons (Andrews et al., 2008; Liu et al., 2012). Similarly, in the context that our animals were fed ad libitum for our studies, and AgRP neuron responsiveness is known to be influenced by feeding state (Chen et al., 2015), we found that the magnitude of our cold-evoked calcium activity in AgRP neurons is less than what is observed following either fasting or ghrelin administration (Chen et al., 2015).”

Revised Materials and methods text: “After baseline correction, dF/F (%) was calculated as individual fluorescence intensity measurements relative to median fluorescence of entire session for 470nm channel. Averaged dF/F (%) for each temperature (30 or 14°C) were limited to the 10-min period either before or after the 1-min ramp either between the 14°C to 30°C transition or the 30°C to 14°C transition.”

2) How is the AgRP neuron inhibition quantified? Is the Fos induced by cold completely gone?a) This may also speak to why heat production is still intact – what if only half of the AgRP neuron activity was blocked?

The inhibitory DREADD viral construct used in this paper was generously donated by Dr. Larry Zweifel (University of Washington) and has been extensively validated and utilized in four publications from his group (M Soden, et al., Cell Rep 2016, E Carlson, et al., Biol Psych 2018, S Miller, et al., Nat Neuro 2019, Y Jo, et al., Nat Comm. 2020), although these publications did not target AgRP neurons. Therefore, to demonstrate our ability to sufficiently inhibit AgRP neurons using DREADDs, we have included additional data showing that in AgRP-IRES-Cre mice that received the inhibitory DREADD virus, a pre-treatment injection of CNO attenuated cold-induced activation of Fos in AgRP neurons relative to saline-treated controls (Saline: 7.2+/-0.1% vs. CNO: 14.9+/-.85%, p = 0.006, unpaired student’s t-test) (Figure 5—figure supplement 1).

Revised Results text: “To test this hypothesis, we utilized a chemogenetic approach in which AgRP-IRES-Cre mice received bilateral microinjections into the ARC of an AAV construct containing a Cre-dependent cassette encoding the inhibitory designer receptor activated by a designer drug (DREADD), hM4Di:eYFP (Sanford et al., 2017) or a fluorescent reporter control alone (mCherry) (Figure 5A). […] To further validate the ability of the inhibitory DREADD to reduce AgRP neuron responsiveness, we found that pre-treatment injection of CNO in mice expressing the inhibitory DREADD in AgRP neurons attenuated cold-induced induction of Fos in AgRP neurons relative to saline-treated controls (Figure 5—figure supplement 1).”

Other comments:3) A paradox is presented here in regards to published literature. Krashes et al., 2011 paper shows AgRP neuron activation decreases energy expenditure. Yet, these neurons are activated in the cold as an animal increase its energy expenditure. Discuss of this is warranted.

We thank the reviewer for raising this important point. Among potential explanations for this paradox is that AgRP neurons are not a homogeneous group and that subsets involved in cold-responsive hyperphagia may be distinct from those controlling energy expenditure. Accordingly, it is possible that during fasting, both subsets are activated, whereas during cold exposure the latter subset is not. This interpretation is consistent with data presented in our manuscript showing that cold exposure activates only a small subset of AgRP neurons (~25%; Figure 3F), whereas fasting appears to activate most if not all AgRP neurons. We therefore interpret our finding that although AgRP neuron silencing prevents cold-induced hyperphagia, it does not prevent the cold-induced increase of EE (in Figure 5I) as evidence that the two responses involve distinct subsets of these neurons. Thus, our data support a model in which thermoregulatory neurocircuits that mount the feeding response to cold exposure are distinct from those that induce thermogenesis, and that the former involves only the feeding-relevant subset of AgRP neurons, and not those capable of suppressing EE. A related point is that physiological and behavioral responses induced by AgRP neuron activation is known to vary depending on the availability of food; during fasting, food is not available by definition, whereas in our studies, food is available ad libitum during cold exposure. We have added text to address this issue (please also refer to point #1 and point #6).

Revised Discussion text: “Our findings further show that chemogenetic inhibition of AgRP neurons selectively blocks cold-induced hyperphagia without impacting cold-induced increases of energy expenditure, respiratory quotient or ambulatory activity, suggesting cold-induced AgRP neuron activation is required for the feeding, but not the thermogenic response to this challenge. […] Future studies to identify the downstream projection sites and determine the contribution of AgRP, NPY, or the inhibitory neurotransmitter, GABA, to cold-induced feeding responses (Krashes et al., 2013) are a priority.”

4) "Instead, the data are more consistent with a model in which AgRP neuron activation contributes to cold-induced hyperphagia."I understand that Agrp and Npy expression are upregulated in hunger, but is this tied uniquely to AgRP neuron activation? Has that experiment been done to enhance this hypothesis?

While our findings demonstrate that activation of AgRP neurons is required for intact feeding responses to cold exposure, whether this effect is mediated via AgRP, NPY, or GABA (each of which can be released from these neurons) remains to be established. With this said, however, previous evidence suggests that whereas release of GABA or NPY is required for the rapid stimulation of feeding, the feeding effects of AgRP (via Mc4r antagonism) occur over a longer period following a delayed onset (Krashes et al., 2013). We agree that future studies are warranted to determine the relative roles of AgRP, NPY, and GABA as mediators of the adaptive hyperphagic response to cold exposure.

5) Figure 3A-C is never called out in text

We have edited the text to correct this omission.

Edited Results text: “We then quantified the total number of *Agrp*+ (GFP), Fos+, and Fos+/AgRP+ across the entire rostral to caudal ARC (Figure 3A) with representative images from the center of this range presented in Figure 3B. Although the total number of AgRP+ neurons was comparable in all mice tested (Figure 3C), relative to mice maintained at 22°C, we found that both the total number of Fos+ cells in the ARC (Figure 3D) and, specifically, the number of Fos+ AgRP neurons (Figure 3E-F) was increased within 90-min of cold-exposure onset.”

6) It is not clear from text/figure that 3F means that ~25% of AGRP neurons have Fos after cold exposure. Is that true? And what does that mean? Especially in light of the photometry data.

The reviewer makes an excellent observation that we have further emphasized and clarified in the Results. As described in both points #1 and #3 above, we interpret the finding that only ~25% of AgRP neurons are Fos+ after cold exposure to suggest that, unlike fasting, cold exposure activates only a subset of AgRP neurons. This interpretation is consistent with our finding that the effect of chemogenetic AgRP neuron inhibition to block cold-induced hyperphagia occurs despite no effect on energy expenditure (Figure 5). Based on these considerations, we hypothesize that while subpopulations of AgRP neurons exist with distinct downstream projection sites that mediate opposing effects on energy intake and energy expenditure, only the subset that stimulates feeding is activated during cold exposure. Future studies to identify and more fully characterize AgRP neuronal subsets and their downstream targets are a priority.

A predicted consequence of this hypothesis is that the magnitude of activity change in AgRP neurons (as determined using FP) in response to cold exposure will be less than occurs during fasting or other challenges that activate AgRP neurons more broadly, such as ghrelin administration, and our data are consistent with this prediction (Please refer to point #1).

We have added text to address this issue (Please refer to point #1 and point #3 for revised Discussion text).

7) What do the authors make of the large number of non AgRP neurons in the arcuate that express Fos? (Figure 3—figure supplement 2).

We thank the reviewer for pointing this out, and additional studies are needed to identify these non-AgRP cold-responsive ARC neuronal subsets. Among the populations of the ARC capable of driving feeding are tyrosine hydroxylase (TH)-expressing neurons (ARC^TH^) (X Zhang, J Neuro, 2015 and Nat Neuro, 2016). To shed additional light on their possible role in cold-induced hyperphagia, we performed an additional qRT-PCR experiment to test for increases in *Th* expression in the ARC with cold using RNA from the study in Figure 1. However, we found no difference in *Th* expression between 14°C and 22°C exposed groups (data not included in manuscript). Additional research is warranted to identify and functionally characterize these non-AgRP, cold-responsive neurons.

The gray shaded area, or bar, represents the transition between temperatures achieved using a Peltier cooler (set to 60 seconds). The figure has been updated such that the gray bar is labeled “ramp” and the Figure 4 legend has been edited to improve clarity:

Edited Figure 4 legend: “(A) Representative diagram of fiber photometry with fiber placement at ARC. (B) Unilateral GCaMP6s expression in AgRP neurons. […] Student’s paired t-test, **p<0.01, *p<0.05.”